# NEAR-OPTIMAL CONVERGENCE
# OF ACCELERATED GRADIENT METHODS
# UNDER GENERALIZED AND $(L_0, L_1)$–SMOOTHNESS

## ABSTRACT

We study first-order methods for convex optimization problems with functions $f$ satisfying the recently proposed $\ell$-smoothness condition $\left\|\nabla^2 f(x)\right\| \le \ell\left(\left\|\nabla f(x)\right\|\right)$, which generalizes the $L$–smoothness and $(L_0, L_1)$–smoothness. While accelerated gradient descent (AGD) is known to reach the optimal complexity $\mathcal{O}(\sqrt{L}R/\sqrt{\varepsilon})$ under $L$–smoothness, where $\varepsilon$ is an error tolerance and $R$ is the distance between a starting and an optimal point, existing extensions to $\ell$–smoothness either incur extra dependence on the initial gradient, suffer exponential factors in $L_1 R$, or require costly auxiliary sub-routines, leaving open whether an AGD-type $\mathcal{O}(\sqrt{\ell(0)}R/\sqrt{\varepsilon})$ rate is possible for small–$\varepsilon$, even in the $(L_0, L_1)$-smoothness case. We resolve this open question. Developing new proof techniques, we achieve $\mathcal{O}(\sqrt{\ell(0)}R/\sqrt{\varepsilon})$ oracle complexity for small–$\varepsilon$ and virtually any $\ell$. For instance, for $(L_0, L_1)$-smoothness, our bound $\mathcal{O}(\sqrt{L_0}R/\sqrt{\varepsilon})$ is provably optimal in the small-$\varepsilon$ regime and removes all non-constant multiplicative factors present in prior accelerated algorithms.

## 1 INTRODUCTION

We focus on optimization problems

$$\min_{x \in \mathbb{R}^d} f(x), \tag{1}$$

where $f : \mathbb{R}^d \to \mathbb{R} \cup \{\infty\}$ is a convex function. We aim to find an $\varepsilon$-solution, $\bar{x} \in \mathbb{R}^d$, such that $f(\bar{x}) - \inf_{x \in \mathbb{R}^d} f(x) \le \varepsilon$. We define $\mathcal{X} = \left\{x \in \mathbb{R}^d \mid f(x) < \infty\right\}$, and assume that $\mathcal{X}$ is an open and $d$–dimensional convex set, $f$ is smooth on $\mathcal{X}$, and continuous on the closure of $\mathcal{X}$. We define $R := \left\|x^0 - x^*\right\|$, where $x^0 \in \mathcal{X}$ is a starting point of numerical methods.

Under the $L$–smoothness assumption, i.e., $\|\nabla f(x) - \nabla f(y)\| \le L\|x - y\|$ or $\left\|\nabla^2 f(x)\right\| \le L$ for all $x, y \in \mathcal{X}$, the problem is well studied. In particular, it is known that one can find an $\varepsilon$-solution after $\mathcal{O}\left(\sqrt{L}R/\sqrt{\varepsilon}\right)$ gradient calls using the fast/accelerated gradient descent method (AGD) by Nesterov (1983), which is also optimal (Nemirovskij & Yudin, 1983; Nesterov, 2018). This result improves the oracle complexity $\mathcal{O}\left(LR^2/\varepsilon\right)$ (# of gradient calculations) of gradient descent (GD).

In this work, we investigate the modern $\ell$–smoothness assumption (Li et al., 2024a), which states that $\left\|\nabla^2 f(x)\right\| \le \ell(\|\nabla f(x)\|)$ for all $x \in \mathcal{X}$ (see Assumption 2.1), where $\ell$ is any non-decreasing, positive, locally Lipschitz function. This generalizes the classical $L$–smoothness assumption, which corresponds to the special case $\ell(s) = L$. An important example of this framework is the $(L_0, L_1)$–smoothness condition (Zhang et al., 2020), obtained by setting $\ell(s) = L_0 + L_1 s$, which yields $\left\|\nabla^2 f(x)\right\| \le L_0 + L_1\|\nabla f(x)\|$ for all $x \in \mathcal{X}$.

There are many functions that are captured by $\ell$–smoothness but not by $L$–smoothness. For instance, $f(x) = x^p$ for $p > 2$, $f(x) = e^x$, and $f(x) = -\log x$ all satisfy $\ell$–smoothness (with a proper $\ell$) but violate the standard $L$–smoothness condition (Li et al., 2024a). Moreover, there is growing evidence that $\ell$–smoothness is a more appropriate assumption for modern machine learning problems (Zhang et al., 2020; Chen et al., 2023; Cooper, 2024; Tyurin, 2025).

Table 1: Convergence rates for various AGD methods for small error tolerance $\varepsilon$ up to constant factors (in the case of $(L_0, L_1)$-Smoothness, the comparison is valid at least for all $\varepsilon \leq L_0/L_1^4 R^2$). Abbreviations: $R := \|x^0 - x^*\|$, $\varepsilon$ = error tolerance, $x^0$ is a starting point, $\Delta := f(x^0) - f(x^*)$, $M_{\bar{R}}$ is defined in Theorem 5.1.

| Setting | Oracle Complexity | References | Required Parameters |
|---|---|---|---|
| $L$–Smoothness | $\frac{\sqrt{L}R}{\sqrt{\varepsilon}}$ | (Nesterov, 1983) | $L$ |
| $(L_0, L_1)$–Smoothness | $\frac{\sqrt{L_0 + L_1\|\nabla f(x^0)\|}R}{\sqrt{\varepsilon}}$ | (Li et al., 2024a) | $L_0, L_1, R, \Delta$ |
| | $\exp(L_1 R) \times \frac{\sqrt{L_0}R}{\sqrt{\varepsilon}}$ | (Gorbunov et al., 2025) | $L_0, L_1$ |
| | $\nu \times \frac{\sqrt{L_0}R}{\sqrt{\varepsilon}}$, where $\nu$ is not a universal constant and may depend on parameters of $f, \varepsilon$, and $R$ | (Vankov et al., 2024) | $L_0, L_1$, params for auxiliary problem (e.g., # of inner iterations) |
| | $\frac{\sqrt{L_0}R}{\sqrt{\varepsilon}}$ | Sec. 3.1, 4.1, or Thm. 4.3 (**new**) | $L_0, L_1, R, \Delta$ (semi-adaptive to $R, \Delta$) |
| General result with any $\ell$ | $\frac{\sqrt{\ell(\|\nabla f(x^0)\|)}R}{\sqrt{\varepsilon}}$ | (Li et al., 2024a) | $L_0, L_1, R, \Delta$ |
| | $\frac{\sqrt{\ell(0)}R}{\sqrt{\varepsilon}}$ | Corollary 5.3 (**new**) | $L_0, L_1, R, \Delta, M_{\bar{R}}$ |

Despite the recent significant interest in $\ell$–smoothness, to the best of our knowledge, one important *open problem* remains:

> Under $\ell$–smoothness and $(L_0, L_1)$–smoothness, for a small $\varepsilon$, is it possible to design a method with oracle complexity $\mathcal{O}\left(\sqrt{\ell(0)}R/\sqrt{\varepsilon}\right)$ and $\mathcal{O}\left(\sqrt{L_0}R/\sqrt{\varepsilon}\right)$, respectively?

In this work, using new proof techniques, we provide an *affirmative answer* to this question by developing new approaches that work for all $\varepsilon > 0$ and achieve the optimal complexity under $(L_0, L_1)$–smoothness for small $\varepsilon$.

## 1.1 RELATED WORK

**Nonconvex optimization with $(L_0, L_1)$–smoothness.** While we focus on convex problems, we now recall the modern results in the non-convex setting. Zhang et al. (2020) is the seminal work that considers $(L_0, L_1)$–smoothness. They developed a clipped version of GD that finds an $\varepsilon$–stationary point after $\mathcal{O}\left(L_0\Delta/\varepsilon + L_1^2\Delta/L_0\right)$ iterations[1]. There are many subsequent works on $(L_0, L_1)$–smoothness, including (Crawshaw et al., 2022; Chen et al., 2023; Wang et al., 2023; Koloskova et al., 2023; Li et al., 2024a;b; Hübler et al., 2024; Vankov et al., 2024). Under $(L_0, L_1)$–smoothness, the state-of-the-art theoretical oracle complexity $\mathcal{O}\left(L_0\Delta/\varepsilon + L_1\Delta/\sqrt{\varepsilon}\right)$ was proved by Vankov et al. (2024).

**Nonconvex optimization with $\ell$–smoothness.** The paper by Li et al. (2024a) is the seminal work that introduces the $\ell$–smoothness assumption. In their version of GD, the result depends on $\ell(\|\nabla f(x^0)\|)/\varepsilon$ and requires $\ell$ to grow more slowly than $s^2$. Subsequently, Tyurin (2025) improved their oracle complexity and provided the current state-of-the-art complexity. For instance, under $(\rho, L_0, L_1)$–smoothness, i.e., $\|\nabla^2 f(x)\| \leq L_0 + L_1\|\nabla f(x)\|^\rho$ for all $x \in \mathcal{X}$, Tyurin (2025) guarantee $L_0\Delta/\varepsilon + L_1\Delta/\varepsilon^{(2-\rho)/2}$ instead of $(L_0\Delta + L_1\|\nabla f(x^0)\|^\rho\Delta)/\varepsilon$ from Li et al. (2024a) when $0 \leq \rho \leq 2$.

**Convex optimization with $(L_0, L_1)$–smoothness and $\ell$–smoothness.** Under the $(L_0, L_1)$–smoothness assumption, convex problems were considered in (Koloskova et al., 2023; Li et al., 2024a; Takezawa et al., 2024). Gorbunov et al. (2025); Vankov et al. (2024) concurrently obtained

---

[1] An $\varepsilon$–stationary point is a point $\bar{x}$ such that $\|\nabla f(\bar{x})\|^2 \leq \varepsilon$; $\Delta := f(x^0) - f^*$, where $x^0$ is a starting point of numerical methods.

the oracle complexity $\mathcal{O}\left(L_0 R^2/\varepsilon + L_1^2 R^2\right)$. Then, the non-dominant term $L_1^2 R^2$ was improved to $L_0 R^2/\varepsilon + \min\left\{L_1 \Delta^{1/2} R/\varepsilon^{1/2}, L_1^2 R^2, L_1\|\nabla f(x^0)\|R^2/\varepsilon\right\}$ by Tyurin (2025). Lobanov et al. (2024) also analyzed the possibility of improving $L_1^2 R^2$ in the region where the gradient of $f$ is large. The $\ell$–smoothness assumption in the contexts of online learning and mirror descent was considered in (Xie et al., 2024; Yu et al., 2025).

**Accelerated convex optimization.** The aforementioned results were derived using non-accelerated gradient descent methods. Under $(L_0, L_1)$–smoothness, accelerated variants of GD were studied by Li et al. (2024a); Gorbunov et al. (2025); Vankov et al. (2024). However, for small $\varepsilon$, the approach of Gorbunov et al. (2025) leads to the complexity $\exp(L_1 R)\sqrt{L_0}R/\sqrt{\varepsilon}$ (up to constant factors), with an exponential dependence on $L_1$ and $R$, while the method proposed by Vankov et al. (2024) requires solving an auxiliary one-dimensional optimization problem at each iteration, leading to the oracle complexity $\mathcal{O}\left(\nu \times \sqrt{L_0}R/\sqrt{\varepsilon}\right)$, where $\nu$ is a non-constant multiplicative factor arising from solving the auxiliary problem. In the context of the $\ell$–smoothness assumption, Li et al. (2024a) established a complexity bound of $\mathcal{O}(\sqrt{\ell(\|\nabla f(x^0)\|)}R/\sqrt{\varepsilon})$. The current state-of-the-art accelerated methods leave open the question of whether it is possible to achieve the oracle complexities $\mathcal{O}\left(\sqrt{L_0}R/\sqrt{\varepsilon}\right)$ and $\mathcal{O}\left(\sqrt{\ell(0)}R/\sqrt{\varepsilon}\right)$ when $\varepsilon$ is small.

## 1.2 Contributions

We develop new proof techniques to analyze Algorithms 1 and 2, which, to the best of our knowledge, achieve for the first time oracle complexities of $\mathcal{O}\left(\sqrt{\ell(0)}R/\sqrt{\varepsilon}\right)$ and $\mathcal{O}\left(\sqrt{L_0}R/\sqrt{\varepsilon}\right)$ for small $\varepsilon$, under $\ell$–smoothness and $(L_0, L_1)$–smoothness, respectively. These results represent a significant improvement over previous works (Li et al., 2024a; Gorbunov et al., 2025; Vankov et al., 2024) (Table 1). Moreover, our bound under $(L_0, L_1)$–smoothness is optimal in the small-$\varepsilon$ regime.

We begin in Section 3, which establishes the $\mathcal{O}\left(\sqrt{\ell(0)}R/\sqrt{\varepsilon}\right)$ rate for small $\varepsilon$ with subquadratic and quadratic $\ell$. In Section 4, we present Algorithm 2, which is more robust to input parameters and achieves an improved rate in the non-dominant terms, at least in the case of $(L_0, L_1)$–smoothness. Finally, in Section 5, we show that Algorithm 1 attains the $\mathcal{O}\left(\sqrt{\ell(0)}R/\sqrt{\varepsilon}\right)$ rate (for small $\varepsilon$) for all non-decreasing positive locally Lipschitz $\ell$.

## 2 Preliminaries

**Notations:** $\mathbb{R}_+ := [0, \infty)$; $\mathbb{N} := \{1, 2, \dots\}$; $\|x\|$ denotes the standard Euclidean norm for all $x \in \mathbb{R}^d$; $\langle x, y \rangle = \sum_{i=1}^d x_i y_i$ denotes the standard dot product; $\|A\|$ denotes the standard spectral norm for all $A \in \mathbb{R}^{d \times d}$; $g = \mathcal{O}(f)$ : there exists $C > 0$ such that $g(z) \leq C \times f(z)$ for all $z \in \mathcal{Z}$; $g = \Omega(f)$ : there exists $C > 0$ such that $g(z) \geq C \times f(z)$ for all $z \in \mathcal{Z}$; $g \simeq h$ : $g$ and $h$ are equal up to a universal positive constant; $\text{Proj}_{\bar{\mathcal{X}}}(x)$ denotes the standard Euclidean projection of $x$ onto the convex closed set $\bar{\mathcal{X}}$.

We consider the following assumption (Li et al., 2024a):

**Assumption 2.1.** A function $f : \mathbb{R}^d \to \mathbb{R} \cup \{\infty\}$ is $\ell$–smooth if $f$ is twice differentiable on $\mathcal{X}$, $f$ is continuous on the closure of $\mathcal{X}$, and there exists a *non-decreasing positive locally Lipschitz* function $\ell : [0, \infty) \to (0, \infty)$ such that

$$\left\|\nabla^2 f(x)\right\| \leq \ell(\|\nabla f(x)\|) \tag{2}$$

for all $x \in \mathcal{X}$.

The assumption includes $L$–smoothness when $\ell(s) = L$, $(L_0, L_1)$–smoothness when $\ell(s) = L_0 + L_1 s$, and $(\rho, L_0, L_1)$–smoothness, i.e., $\left\|\nabla^2 f(x)\right\| \leq L_0 + L_1 \|\nabla f(x)\|^\rho$ for all $x \in \mathcal{X}$, when $\ell(s) = L_0 + L_1 s^\rho$, where $L, L_0, L_1, \rho \geq 0$ are some fixed constants. While Assumption 2.1 requires twice differentiability, the main theorems and algorithms do not directly rely on it. Let us recall the following lemma, which follows from Assumption 2.1:

**Lemma 2.2** (Tyurin (2025)). *For all $x, y \in \mathcal{X}$ such that $\|y - x\| \in [0, q_{\max}(\|\nabla f(x)\|))$, if $f$ is $\ell$–smooth (Assumption 2.1), then*

$$\|\nabla f(y) - \nabla f(x)\| \leq q^{-1}(\|y - x\| ; \|\nabla f(x)\|), \tag{3}$$

---

**Algorithm 1** Accelerated Gradient Descent (AGD) with $\ell$-Smoothness

---

1: **Input:** starting point $x^0 \in \mathcal{X}$, function $\ell$ from Assumption 2.1, parameters $\delta$ and $\bar{R}$
2: Starting from $x^0$, run GD from (Tyurin, 2025) until $f(\bar{x}) - f(x^*) \leq \delta/2$,
   where $\bar{x}$ is the output point of GD
3: Init $y^0 = u^0 = \bar{x}$
4: Set $\Gamma_0 = \delta/\bar{R}^2$
5: Set $\gamma = 1/(2\ell(0))$
6: **for** $k = 0, 1, \ldots$ **do**
7: $\quad \alpha_k = \sqrt{\gamma \Gamma_k}$
8: $\quad y^{k+1} = \frac{1}{1+\alpha_k} y^k + \frac{\alpha_k}{1+\alpha_k} u^k - \frac{\gamma}{1+\alpha_k} \nabla f(y^k)$
9: $\quad u^{k+1} = \mathrm{Proj}_{\bar{\mathcal{X}}} \left( u^k - \frac{\alpha_k}{\Gamma_k} \nabla f(y^{k+1}) \right)$ $\qquad\qquad$ ($\bar{\mathcal{X}}$ is the closure of $\mathcal{X}$)
10: $\quad \Gamma_{k+1} = \Gamma_k/(1 + \alpha_k)$
11: **end for**

---

where $q(s; a) := \int_0^s \frac{dv}{\ell(a+v)}$, $q^{-1}$ is the inverse of $q$ with respect to $s$, and $q_{\max}(a) := \int_0^\infty \frac{dv}{\ell(a+v)}$.

Not requiring twice differentiability, we can assume that (3) holds instead of (2). The main reason why we start with (2) is because it is arguably more interpretable. Next, we assume the convexity of $f$:

**Assumption 2.3.** A function $f : \mathbb{R}^d \to \mathbb{R} \cup \{\infty\}$ is convex and attains the minimum at a (non-unique) $x^* \in \mathbb{R}^d$. We define $R := \|x^0 - x^*\|$, where $x^0$ is a starting point of numerical methods.

In the theoretical analysis and proofs, it is useful to define the $\psi$–function:

**Definition 2.4** ($\psi$ and $\psi^{-1}$ functions). Let Assumption 2.1 hold. We define the function $\psi : \mathbb{R}_+ \to \mathbb{R}_+$ such that $\psi(x) = \frac{x^2}{2\ell(4x)}$, and $\psi^{-1} : [0, \psi(\Delta_{\max})) \to [0, \Delta_{\max})$ as its (standard) inverse, where $\Delta_{\max} \in (0, \infty]$ is the largest constant such that $\psi$ is strictly increasing on[2] $[0, \Delta_{\max})$.

## 3 SUBQUADRATIC AND QUADRATIC GROWTH OF $\ell$

We are ready to present our first result. Consider Algorithm 1, which consists of two phases: first, we run (non-accelerated) GD, and then we run an accelerated version of GD. Later, we will present Algorithm 2, which avoids the first phase. We first state the convergence rate of Algorithm 1 and then discuss and explain it in more detail. We begin by stating a standard result from the theory of accelerated methods (Nesterov, 2018; Lan, 2020; Stonyakin et al., 2021) concerning auxiliary sequences, which control convergence rates:

**Theorem 3.1.** *For any $\Gamma_0 > 0$ and $\gamma \geq 0$, let $\alpha_k \geq \sqrt{\gamma \Gamma_k}$ and $\Gamma_{k+1} = \Gamma_k/(1 + \alpha_k)$ for all $k \geq 0$. Then, $\Gamma_{k+1} \leq \frac{9}{\gamma(k+1-\bar{k})^2}$ for all $k \geq \bar{k} := \max\left\{1 + \frac{1}{2}\log_{3/2}\left(\frac{\gamma\Gamma_0}{4}\right), 0\right\}$.*

The following result provides the convergence rate of Algorithm 1 for $\ell$ such that $\psi(x) = \frac{x^2}{2\ell(4x)}$ is strictly increasing, which holds, for instance, under $(L_0, L_1)$–smoothness.

**Theorem 3.2.** *Suppose that Assumptions 2.1 and 2.3 hold. Let $\psi : \mathbb{R}_+ \to \mathbb{R}_+$ such that $\psi(x) = \frac{x^2}{2\ell(4x)}$ be strictly increasing. Then Algorithm 1 guarantees that*

$$f(y^{k+1}) - f(x^*) \leq \Gamma_{k+1}\bar{R}^2 \leq \frac{18\ell(0)\bar{R}^2}{(k + 1 - \bar{k})^2} \tag{4}$$

*for all $k \geq \bar{k} := \max\left\{1 + \frac{1}{2}\log_{3/2}\left(\frac{\Gamma_0}{8\ell(0)}\right), 0\right\}$ with any $\delta \in (0, \infty]$ such that $\ell\left(8\sqrt{\delta\ell(0)}\right) \leq 2\ell(0)$ and any $\bar{R} \geq R := \|x^0 - x^*\|$.*

---

[2]$\Delta_{\max} > 0$ due to Lemma B.4.

The theorem establishes the desired $1/k^2$ convergence rate of accelerated methods. However, the method enters this regime only after running the GD method and after the initial $\bar{k}$ steps of the accelerated steps. The main and final result, which captures the total oracle complexity, is presented below.

**Theorem 3.3.** *Consider the assumptions and results of Theorem 3.2. The oracle complexity (i.e., the number of gradient calls) required to find an $\varepsilon$–solution is*

$$\frac{5\sqrt{\ell(0)}\bar{R}}{\sqrt{\varepsilon}} + k(\delta), \tag{5}$$

*for all $\delta \geq 0$ such that $\ell\left(8\sqrt{\delta\ell(0)}\right) \leq 2\ell(0)$, where $k(\delta) := \max\left\{1 + \frac{1}{2}\log_{3/2}\left(\frac{\delta}{8\ell(0)\bar{R}^2}\right), 0\right\} + k_{\mathrm{GD}}(\delta)$, $k_{\mathrm{GD}}(\delta)$ is the oracle complexity of GD for finding a point $\bar{x}$ such that $f(\bar{x}) - f(x^*) \leq \delta/2$.*

**Corollary 3.4.** *In Theorem 3.3, minimizing over $\delta$ and taking $\bar{R} = R := \left\|x^0 - x^*\right\|$, the oracle complexity is*

$$\frac{5\sqrt{\ell(0)}R}{\sqrt{\varepsilon}} + \underbrace{\min_{\delta \geq 0 \,:\, \ell\left(8\sqrt{\delta\ell(0)}\right) \leq 2\ell(0)} k(\delta)}_{\text{does not depend on } \varepsilon}. \tag{6}$$

### 3.1 EXAMPLE: $(L_0, L_1)$–SMOOTHNESS

We now consider an example and apply the result for $(L_0, L_1)$–smooth functions. In this case, $\ell(s) = L_0 + L_1 s$. First, we need to find the proper set of $\delta$ from Theorem 3.2: $\ell(8\sqrt{\delta\ell(0)}) \leq 2\ell(0) \Leftrightarrow L_0 + L_1(8\sqrt{\delta L_0}) \leq 2L_0 \Leftrightarrow \delta \leq L_0/(64L_1^2)$. Second, we need to find $k_{\mathrm{GD}}(\delta)$. Using Table 2 from (Tyurin, 2025), or the results by Gorbunov et al. (2025); Vankov et al. (2024), $k_{\mathrm{GD}}(\delta) = \mathcal{O}\left(L_0 R^2/\delta + \min\left\{L_1\Delta^{1/2}R/\delta^{1/2}, L_1^2 R^2, L_1\left\|\nabla f(x^0)\right\|R^2/\delta\right\}\right) = \mathcal{O}\left(\frac{L_0 R^2}{\delta}\right) = \mathcal{O}\left(\frac{L_0\bar{R}^2}{\delta}\right)$ for all $\delta \leq L_0/(64L_1^2)$. Substituting to (5), we get the total oracle complexity

$$\mathcal{O}\left(\frac{\sqrt{L_0}\bar{R}}{\sqrt{\varepsilon}} + \min_{0 \leq \delta \leq L_0/(64L_1^2)}\left[\max\left\{\log\left(\frac{\delta}{L_0\bar{R}^2}\right), 0\right\} + \frac{L_0\bar{R}^2}{\delta}\right]\right), \tag{7}$$

Taking $\delta = \min\{L_0/(64L_1^2), (L_0\bar{R}^2)/64\}$ (which might not be the optimal choice, but a sufficient choice to show that the first term dominates if $\varepsilon$ is small), we get

$$(7) = \mathcal{O}\left(\frac{\sqrt{L_0}\bar{R}}{\sqrt{\varepsilon}} + L_1^2\bar{R}^2\right) = \mathcal{O}\left(\frac{\sqrt{L_0}R}{\sqrt{\varepsilon}} + L_1^2 R^2\right), \tag{8}$$

where we choose $\bar{R} = R$. Unlike Li et al. (2024a); Gorbunov et al. (2025); Vankov et al. (2024), we get $\mathcal{O}(\sqrt{L_0}R/\sqrt{\varepsilon})$ for small $\varepsilon$. Moreover, this complexity is optimal (Nemirovskij & Yudin, 1983; Nesterov, 2018) for small $\varepsilon$ in the sense that for any $L_0 > 0$ and $L_1 \geq 0$, it is possible to find an $(L_0, L_1)$–smooth function (the $(L_0, 0)$–smooth function from Section 2.1.2 of (Nesterov, 2018)) such that the required number of oracle calls is $\Omega(\sqrt{L_0}R/\sqrt{\varepsilon})$ for small $\varepsilon$.

One can repeat these steps for any $\ell$ such that $\psi$ is strictly increasing. Nevertheless, even without these derivations, we establish the total oracle complexity $\mathcal{O}(\sqrt{\ell(0)}R/\sqrt{\varepsilon})$ in (6) for small $\varepsilon$.

### 3.2 DISCUSSION

The closest work to the complexity $\mathcal{O}\left(\sqrt{L_0}R/\sqrt{\varepsilon}\right)$, when $\varepsilon$ is small, is (Vankov et al., 2024). Using the same idea as in (Vankov et al., 2024), in Algorithm 1, we run GD until $f(\bar{x}) - f(x^*) \leq \delta/2$. However, the next steps and proof techniques are new. Using the "warm-start" point $\bar{x}$, it becomes easier for Algorithm 1 to run accelerated steps because we take $\delta$ such that $\ell(4\left\|\nabla f(y^0)\right\|) \leq 2\ell(0)$ (Lemma E.1), meaning that we start from the region where the local smoothness constant is almost $\ell(0)$. The main challenge is to ensure that the next points $y^k$ of Algorithm 1 never leave this region. To ensure that, using the method from (Nesterov et al., 2021), Vankov et al. (2024) utilize the monotonicity of their accelerated method and the fact that their points do not leave the region with

small smoothness. However, it is not for free and requires $\nu$ extra oracle calls in each iteration, where $\nu$ is not a universal constant and depends on the parameters of $f$ leading to a suboptimal complexity.

In contrast, our method follows the standard approach, where only one gradient is computed per iteration. We use the version of the accelerated method from (Wei & Chen, 2025)[Section D.2], with some minor but important modifications. The method itself is very similar to the one from (Allen-Zhu & Orecchia, 2014), for instance. However, the proof technique is very different, which is the main reason we focus on Algorithm 1. While for $L$–smooth functions the proof technique from (Wei & Chen, 2025) does not offer any advantages over, for example, (Nesterov, 1983) because the result in (Nesterov, 1983) is optimal. In the case of functions with generalized smoothness, it becomes particularly useful, as shown in the following section.

### 3.3 PROOF SKETCH

As in most proofs, we define the Lyapunov function $V_k := f(y^k) - f(x^*) + \frac{\Gamma_k}{2} \left\| u^k - x^* \right\|^2$. The first important observation is that in $V_k$ we use $y^k$, the point where the gradient is actually computed. This is important, and we will see why later.

Using mathematical induction, let us assume that we have run Algorithm 1 up to $k^{\text{th}}$ iteration, $\ell\left(4\left\|\nabla f(y^k)\right\|\right) \le 2\ell(0)$, and $V_k \le \left(\prod_{i=0}^{k-1} \frac{1}{1+\alpha_i}\right) V_0$. We choose $\Gamma_0$ such that $V_0 \le \delta$. The base case with $k = 0$ is true because we run GD until $\ell(4\left\|\nabla f(y^0)\right\|) \le 2\ell(0)$. Now, instead of $k + 1^{\text{th}}$ consider the steps

$$
\begin{aligned}
\alpha_{k,\gamma} &= \sqrt{\gamma \Gamma_k}, \\
y_\gamma^{k+1} &= \frac{1}{1+\alpha_{k,\gamma}} y^k + \frac{\alpha_{k,\gamma}}{1+\alpha_{k,\gamma}} u^k - \frac{\gamma}{1+\alpha_{k,\gamma}} \nabla f(y^k), \\
u_\gamma^{k+1} &= \text{Proj}_{\bar{\mathcal{X}}}\left(u^k - \frac{\alpha_{k,\gamma}}{\Gamma_k} \nabla f(y_\gamma^{k+1})\right), \\
\Gamma_{k+1,\gamma} &= \Gamma_k/(1+\alpha_{k,\gamma}),
\end{aligned}
\tag{9}
$$

where $\gamma$ is a free parameter. These steps are equivalent to $k + 1^{\text{th}}$ iteration when $\gamma = 1/\left(2\ell(0)\right)$. However, we have not proved that we are allowed to use this $\gamma$ yet. For these steps, we can prove a standard descent lemma, Lemma D.1:

$$
\left[(1+\alpha_{k,\gamma})(f(y_\gamma^{k+1}) - f(x^*)) + \frac{(1+\alpha_{k,\gamma})\Gamma_{k+1,\gamma}}{2}\left\|u_\gamma^{k+1} - x^*\right\|^2\right] - V_k
$$

$$
\le \frac{1}{2}\left(\gamma - \frac{1}{\ell(2\left\|\nabla f(y^k)\right\| + \left\|\nabla f(y_\gamma^{k+1})\right\|)}\right)\left\|\nabla f(y_\gamma^{k+1}) - \nabla f(y^k)\right\|^2.
\tag{10}
$$

For now, let us assume that $f$ is $L$–smooth. Then the rest of the proof becomes straightforward. In this case, $\ell(2\left\|\nabla f(y^k)\right\| + \left\|\nabla f(y_\gamma^{k+1})\right\|) = L$, and we can take $\gamma = 1/2L \equiv 1/(2\ell(0))$ to ensure that $(1 + \alpha_k)V_{k+1} \le V_k$ because the first bracket $[\ldots] = (1 + \alpha_k)V_{k+1}$. Then, we should unroll the recursion and use Theorem 3.1 to get the classical $1/k^2$ rate (Nesterov, 1983).

However, under Assumption 2.1, $\ell(2\left\|\nabla f(y^k)\right\| + \left\|\nabla f(y_\gamma^{k+1})\right\|)$ depends on $\left\|\nabla f(y_\gamma^{k+1})\right\|$, and we encounter a "chicken-and-egg" dilemma: in order to choose $\gamma$, we need to know $\left\|\nabla f(y_\gamma^{k+1})\right\|$, which in turn depends on $\gamma$. Our resolution is the following. Let us choose the smallest $\gamma^* \ge 0$ such that

$$
g(\gamma) := \gamma - \frac{1}{\ell(2\left\|\nabla f(y^k)\right\| + \left\|\nabla f(y_\gamma^{k+1})\right\|)} = 0,
$$

which exists and is positive because $g(\gamma)$ is continuous, $g(0) < 0$, and $g(\bar{\gamma}) \ge 0$ for $\bar{\gamma} = \frac{1}{\ell(2\|\nabla f(y^k)\|)}$. It is possible that we are "unlucky" and $\gamma^*$ is very small, leading to a slow convergence rate and preventing us from choosing $\gamma = 1/(2\ell(0))$. Surprisingly, it is possible to show that $\gamma^* \ge 1/(2\ell(0))$. Indeed, using (10), for all $\gamma \le \gamma^*$, we have $f(y_\gamma^{k+1}) - f(x^*) \le V_k \le V_0$. Recall that we choose $\Gamma_0$ such that $V_0 \le \delta$. Thus, $f(y_\gamma^{k+1}) - f(x^*) \le \delta$. This is the key inequality in the proof, which allows us to conclude that the function gap with $y_\gamma^{k+1}$ is bounded, thus justifying the choice of the Lyapunov function.

---

**Algorithm 2** AGD with $\ell$-smoothness and increasing step sizes (without GD pre-running)

---

1: **Input:** starting point $x^0 \in \mathcal{X}$, function $\ell$ from Assumption 2.1, parameters $\Gamma_0$ and $\bar{R}$
2: Init $y^0 = u^0 = x^0$
3: Define $\psi(x) = \frac{x^2}{2\ell(4x)}$          (assume that $\psi$ is invertible on $\mathbb{R}_+$)
4: **for** $k = 0, 1, \ldots$ **do**
5:     $\gamma_k = 1/\ell\left(4\psi^{-1}\left(\Gamma_k \bar{R}^2\right)\right)$
6:     $\alpha_k = \sqrt{\gamma_k \Gamma_k}$
7:     $y^{k+1} = \frac{1}{1+\alpha_k}y^k + \frac{\alpha_k}{1+\alpha_k}u^k - \frac{\gamma_k}{1+\alpha_k}\nabla f(y^k)$
8:     $u^{k+1} = \text{Proj}_{\bar{\mathcal{X}}}\left(u^k - \frac{\alpha_k}{\Gamma_k}\nabla f(y^{k+1})\right)$        ($\bar{\mathcal{X}}$ is the closure of $\mathcal{X}$)
9:     $\Gamma_{k+1} = \Gamma_k/(1+\alpha_k)$
10: **end for**

---

It left to use Lemma E.1, which allows us to bound $\ell(4\|\nabla f(y)\|)$ if we can bound $f(y) - f(x^*) \leq \delta$ for all $y \in \mathcal{X}$. Thus, $\ell\left(4\|\nabla f(y_\gamma^{k+1})\|\right) \leq 2\ell(0)$ for all $\gamma \leq \gamma^*$. Recalling the definition of $\gamma^*$ :

$$\gamma^* = \frac{1}{\ell(2\|\nabla f(y^k)\| + \|\nabla f(y_{\gamma^*}^{k+1})\|)} \geq \frac{1}{\max\{\ell(4\|\nabla f(y^k)\|), \ell(4\|\nabla f(y_{\gamma^*}^{k+1})\|)\}} \geq \frac{1}{2\ell(0)}.$$

Finally, this means that we can take $\gamma = 1/(2\ell(0))$, (9) reduces to the $k + 1^{\text{th}}$ step of Algorithm 1, $\ell\left(4\|\nabla f(y^{k+1})\|\right) \leq 2\ell(0)$, and $V_{k+1} \leq \left(\prod_{i=0}^{k}\frac{1}{1+\alpha_i}\right)V_0$ due to (10). We have proved the next step of mathematical induction and (4).

The way we resolve the "chicken-and-egg" dilemma can be an interesting proof trick in other optimization contexts. Note that our method is not necessarily monotonic, but the proof still allows us to show that the method never leaves the region where the local smoothness constant is almost $\ell(0)$.

# 4 STABILITY WITH RESPECT TO INPUT PARAMETERS AND IMPROVED RATES

While, to the best of our knowledge, Algorithm 1 is the first algorithm with $\mathcal{O}\left(\sqrt{\ell(0)}R/\sqrt{\varepsilon}\right)$ complexity, it has two limitations: it runs GD at the beginning, and it requires a good estimate of $R$ when selecting $\bar{R}$. We resolve these issues in Algorithm 2, which is similar to Algorithm 1, but the former does not run GD at the beginning, uses the step sizes $\gamma_k = 1/\ell\left(4\psi^{-1}\left(\Gamma_k\bar{R}^2\right)\right)$, and requires $\Gamma_0$ as an input.

**Theorem 4.1.** *Suppose that Assumptions 2.1 and 2.3 hold. Let $\psi : \mathbb{R}_+ \to \mathbb{R}_+$ such that $\psi(x) = \frac{x^2}{2\ell(4x)}$ be strictly increasing and $\lim_{x \to \infty} \psi(x) = \infty$. Then Algorithm 2 guarantees that*

$$f(y^{k+1}) - f(x^*) \leq \Gamma_{k+1}R^2$$

*for all $k \geq 0$ with $\Gamma_0 \geq \frac{2(f(x^0) - f(x^*))}{\|x^0 - x^*\|^2}$ and $\bar{R} \geq R$.*

**Theorem 4.2.** *Consider the assumptions and results of Theorem 4.1. The oracle complexity (i.e., the number of gradient calls) required to find an $\varepsilon$–solution is*

$$\frac{5\sqrt{\ell(0)}R}{\sqrt{\varepsilon}} + \underbrace{\max\left\{2 + \log_{3/2}\left(\frac{\Gamma_0}{4\ell(0)}\right), 0\right\} + k_{\text{init}}}_{\text{does not depend on }\varepsilon} \tag{11}$$

*with $\Gamma_0 \geq \frac{2(f(x^0) - f(x^*))}{\|x^0 - x^*\|^2}$, $\bar{R} \geq R$, and $k_{\text{init}}$ being the smallest integer such that*

$$\ell\left(24\sqrt{\frac{\ell\left(4\psi^{-1}\left(\Gamma_0\bar{R}^2\right)\right)\ell(0)\bar{R}^2}{k_{\text{init}}^2}}\right) \leq 2\ell(0).$$

Comparing (11) and (7), one can see that Algorithm 2 is stable with respect to the choice of $\bar{R}$ and $\Gamma_0$. Ideally, it is better to choose $\Gamma_0 = \frac{2(f(x^0) - f(x^*))}{\|x^0 - x^*\|^2}$ and $\bar{R} = R$. However, if we overestimate $\bar{R}$ and $\Gamma_0$, the penalty for this appears in the term that does not depend on $\varepsilon$. In the next section, we consider an example to illustrate this.

## 4.1 EXAMPLE: $(L_0, L_1)$–SMOOTHNESS

To find the oracle complexity, we have to estimate $k_{\text{init}}$. In the case of $(L_0, L_1)$–smoothness, we can find $k_{\text{init}}$ from the equality $L_0 + L_1\sqrt{(L_0 + L_1\psi^{-1}\left(\Gamma_0\bar{R}^2\right))L_0\bar{R}^2/k_{\text{init}}^2} \simeq 2L_0$ (we ignore constants for simplicity), where $\psi^{-1}$ is the inverse of $x^2/(2(L_0 + 4L_1 x))$. If $\Gamma_0\bar{R}^2 \geq L_0/L_1$, then the equality is equivalent to $k_{\text{init}} \simeq \sqrt{L_1^2\bar{R}^2 + L_1^4\Gamma_0\bar{R}^4/L_0}$. Otherwise, $k_{\text{init}} \simeq \sqrt{L_1^2\bar{R}^2 + L_1^3\bar{R}^3\sqrt{\Gamma_0/L_0}}$. Thus, using (11), the total oracle complexity

$$\mathcal{O}\left(\frac{\sqrt{L_0}R}{\sqrt{\varepsilon}} + L_1\bar{R} + L_1^2\bar{R}^2\sqrt{\frac{\Gamma_0}{L_0}} + \max\left\{\log\left(\frac{\Gamma_0}{L_0}\right), 0\right\}\right),\tag{12}$$

where the first term is stable to the choice of $\bar{R}$ and $\Gamma_0$.

## 4.2 SPECIALIZATION FOR $(L_0, L_1)$–SMOOTHNESS

The previous theorems work with any $\ell$ such that $\psi(x) = \frac{x^2}{2\ell(4x)}$ is strictly increasing on $\mathbb{R}_+$ and $\lim_{x\to\infty} \psi(x) = \infty$. It turns out that we can improve (12) and refine Theorem 4.2 in the case of $(L_0, L_1)$–smoothness.

**Theorem 4.3.** *Consider the assumptions and results of Theorem 4.1 with $\ell(s) = L_0 + L_1 s$. The oracle complexity (i.e., the number of gradient calls) required to find an $\varepsilon$–solution is*

$$\mathcal{O}\left(\frac{\sqrt{L_0}R}{\sqrt{\varepsilon}} + \max\left\{L_1\bar{R}\log\left(\min\left\{\frac{L_1^2\bar{R}^2\Gamma_0}{L_0}, \frac{\Gamma_0 R^2}{\varepsilon}\right\}\right), 0\right\} + \max\left\{\log\left(\frac{\Gamma_0}{L_0}\right), 0\right\}\right)\tag{13}$$

*with $\Gamma_0 \geq \frac{2(f(x^0) - f(x^*))}{\|x^0 - x^*\|^2}$ and $\bar{R} \geq R$.*

The non-dominant term in (13) is better than that of (12), and is better than that of (8) when $\Gamma_0 = {}^{2\Delta}/_{R^2}$ and $\bar{R} = R$.

## 4.3 DISCUSSION AND PROOF SKETCH

Unlike Algorithm 1, Algorithm 2 starts from $x^0$ where the initial local smoothness might be large. Nevertheless, the proof follows the proof techniques from Section 3.3 with one important difference: using mathematical induction, we prove that $\left\|\nabla f(y^k)\right\| \leq \psi^{-1}(\Gamma_k\bar{R}^2)$ for all $k \geq 0$. This inequality means that $\left\|\nabla f(y^k)\right\|$ can be bounded by a decreasing sequence, and after several iterations, all $y^k$ satisfy $\ell(4\left\|\nabla f(y^k)\right\|) \leq 2\ell(0)$, allowing us to get $\mathcal{O}(\sqrt{\ell(0)}R/\sqrt{\varepsilon})$ complexity for small–$\varepsilon$.

# 5 SUPERQUADRATIC GROWTH OF $\ell$

In the previous sections, we provided convergence rates under the assumption that $\psi$ is strictly increasing. For instance, the previous theory applies to $(\rho, L_0, L_1)$–smooth functions only if $\rho \leq 2$. For cases where $\psi$ is not necessarily strictly increasing, we can prove the following theorems.

**Theorem 5.1.** *Suppose that Assumptions 2.1 and 2.3 hold. Let $\psi : \mathbb{R}_+ \to \mathbb{R}_+$ such that $\psi(x) = \frac{x^2}{2\ell(4x)}$ be not necessarily strictly increasing. Find the largest $\Delta_{\max} \in (0, \infty]$ such that $\psi$ is strictly increasing on $[0, \Delta_{\max})$. For all $\delta \in [0, \psi(\Delta_{\max}))$, find the unique $\Delta_{\text{left}}(\delta) \in [0, \Delta_{\max})$ and the smallest[3] $\Delta_{\text{right}}(\delta) \in [\Delta_{\max}, \infty]$ such that $\psi(\Delta_{\text{left}}(\delta)) = \delta$ and $\psi(\Delta_{\text{right}}(\delta)) = \delta$. Take any $\delta \in [0, \frac{1}{2}\psi(\Delta_{\max})]$ such that $\ell(4\Delta_{\text{left}}(\delta)) \leq 2\ell(0)$ and $\Delta_{\text{right}}(\delta) \geq 2M_{\bar{R}}$, where[4] $M_{\bar{R}} := \max_{\|x - x^*\| \leq 2\bar{R}} \left\|\nabla f(x)\right\|$. Then Algorithm 1 guarantees that*

$$f(y^{k+1}) - f(x^*) \leq \Gamma_{k+1}\bar{R}^2 \leq \frac{18\ell(0)\bar{R}^2}{\left(k + 1 - \bar{k}\right)^2}$$

---

[3] if the set $\{x \in [\Delta_{\max}, \infty) : \psi(x) = \delta\}$ is empty, then $\Delta_{\text{right}}(\delta) = \infty$

[4] or is it sufficient to find any $M_{\bar{R}}$ such that $M_{\bar{R}} \geq \max_{\|x - x^*\| \leq 2\bar{R}} \|\nabla f(x)\|$.

*for all $k \geq \bar{k} := \max \left\{ 1 + \frac{1}{2} \log_{3/2} \left( \frac{\Gamma_0}{8\ell(0)} \right), 0 \right\}$ with any $\bar{R} \geq \left\| x^0 - x^* \right\|$.*

In order to apply the theorem and algorithm, we first have to find the largest $\Delta_{\max} \in (0, \infty]$ such that $\psi$ is strictly increasing on $[0, \Delta_{\max})$. If $\psi$ is strictly increasing on $\mathbb{R}_+$, then $\Delta_{\max} = \infty$. Next, we should find $\Delta_{\text{left}}(\delta)$ and $\Delta_{\text{right}}(\delta)$ for all $\delta \in [0, \psi(\Delta_{\max}))$. The point $\Delta_{\text{left}}(\delta) \in [0, \Delta_{\max})$ is the solution of $\psi(\Delta_{\text{left}}(\delta)) = \delta$, which exists and is unique for all $\delta \in [0, \psi(\Delta_{\max}))$ because $\psi$ is strictly increasing on $[0, \Delta_{\max})$. Notice that $\psi(x) > \delta$ for all $x \in (\Delta_{\text{left}}(\delta), \Delta_{\max})$. Thus, there are two options: either $\psi(x) > \delta$ for all $x \in (\Delta_{\text{left}}(\delta), \infty)$, and we define $\Delta_{\text{right}}(\delta) = \infty$, or there exists the first moment $\Delta_{\text{right}}(\delta) \in [\Delta_{\max}, \infty)$ when $\psi(\Delta_{\text{right}}(\delta)) = \delta$. In other words, $\Delta_{\text{right}}(\delta)$ is the second time when $\psi$ intersects $\delta$. We define the set of $\delta$ allowed to use in the algorithm:

$$ Q := \left\{ \delta \in [0, \psi(\Delta_{\max})/2] : \ell(4\Delta_{\text{left}}(\delta)) \leq 2\ell(0), \Delta_{\text{right}}(\delta) \geq 2M_{\bar{R}} \right\}. $$

**Theorem 5.2.** *Consider the assumptions and results of Theorem 5.1. The oracle complexity (i.e., the number of gradient calls) required to find an $\varepsilon$–solution is*

$$ \frac{5\sqrt{\ell(0)}\bar{R}}{\sqrt{\varepsilon}} + k(\delta) $$

*for all $\delta \in Q$, where $k(\delta) := \max \left\{ 1 + \frac{1}{2} \log_{3/2} \left( \frac{\delta}{8\ell(0)\bar{R}^2} \right), 0 \right\} + k_{\text{GD}}(\delta)$, $k_{\text{GD}}(\delta)$ is the oracle complexity of GD for finding a point $\bar{x}$ such that $f(\bar{x}) - f(x^*) \leq \delta/2$.*

**Corollary 5.3.** *In Theorem 5.2, minimizing over $\delta$ and taking $\bar{R} = R := \left\| x^0 - x^* \right\|$, the oracle complexity is*

$$ \frac{5\sqrt{\ell(0)}R}{\sqrt{\varepsilon}} + \underbrace{\min_{\delta \in Q} k(\delta)}_{\text{does not depend on } \varepsilon}. \tag{14} $$

In Section E.3.1, we consider an example, $(\rho, L_0, L_1)$–smoothness, to illustrate how to use the theorem, and show that it guarantees a rate of $\sqrt{L_0}R/\sqrt{\varepsilon}$ rate for any $\rho \geq 0$ and a sufficiently small $\varepsilon$. The main observation in (14) is that we obtain the $\sqrt{\ell(0)}R/\sqrt{\varepsilon}$ rate for small $\varepsilon$, given an appropriate or optimal choice of $\delta$ that minimizes $k(\delta)$. The main difference between Theorem 5.2 and Theorem 3.3 is that the rate in Theorem 5.2 depends on $M_{\bar{R}}$ and requires its estimate.

### 5.1 Discussion and proof sketch

In the superquadratic case, we use Algorithm 1 instead of Algorithm 2 because the latter relies on the fact that $\psi$ is invertible on $\mathbb{R}_+$. The former algorithm does not need this and allows us to get the $\sqrt{L_0}R/\sqrt{\varepsilon}$ rate for small–$\varepsilon$. While once again the proof of Theorem 5.2 follows the discussion from Section 3.3, there is one important difference. Since $\psi$ might not be invertible, we cannot conclude that $\left\| \nabla f(y^k) \right\| \leq \psi^{-1}(\delta)$ if $f(y^k) - f(x^*) \leq \delta$. Instead, we can only guarantee that if $f(y^k) - f(x^*) \leq \delta$ and $\delta \in [0, \psi(\Delta_{\max}))$, then either $\left\| \nabla f(y^k) \right\| \leq \Delta_{\text{left}}(\delta)$ or $\left\| \nabla f(y^k) \right\| \geq \Delta_{\text{right}}(\delta)$, where $\Delta_{\max}, \Delta_{\text{left}}(\delta)$, and $\Delta_{\text{right}}(\delta)$ are defined in Section 5. The latter case is "bad" for the analysis. To avoid it, we take $\delta$ such that $\Delta_{\text{right}}(\delta) \geq 2M_{\bar{R}} = \max_{\|x - x^*\| \leq 2\bar{R}} \|\nabla f(x)\|$ and, using mathematical induction, ensure that $\left\| \nabla f(y^k) \right\| \leq M_{\bar{R}}$. To get the last bound, we prove that $y^k$ never leaves the ball $B(x^*, 2\bar{R})$, which requires additional technical steps. Thus, we are left with the "good" case $\left\| \nabla f(y^k) \right\| \leq \Delta_{\text{left}}(\delta)$, which yields $\ell(4 \left\| \nabla f(y^k) \right\|) \leq 2\ell(0)$ for $\delta$ such that $\ell(4\Delta_{\text{left}}(\delta)) \leq 2\ell(0)$.

## 6 Conclusion

While we have achieved a better oracle complexity for small $\varepsilon$, the optimal non-dominant term for large $\varepsilon$, which can improve the terms not depending on $\varepsilon$ in Corollaries 3.4, 5.3 and Theorem 4.2 for $\ell$–smooth functions, remains unclear and require further investigations. Moreover, it would be interesting to extend our results to stochastic and finite-sum settings (Schmidt et al., 2017; Lan, 2020), and develop adaptive versions of the methods than do not depend on $L_0, L_1, R, \Delta$. We leave these directions for future work, which can build on our new insights and algorithms.

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

CONTENTS

# A  EXPERIMENTS

## A.1  COMPARISON WITH GD

We compare GD (Tyurin, 2025) and AGD (Algorithm 2) on the function $f : \mathbb{R}^2 \to \mathbb{R}$ defined as $f(x, y) = e^x + e^{1-x} + \frac{\mu}{2} y^2$, where $\mu = 0.001$. This function is $(3.3 + \mu, 1)$–smooth and has its minimum at $(0.5, 0)$. Starting at $x^0 = (-6, -5)$, and taking $\bar{R} = 100 \gg R$ and $\Gamma_0 = 100 \gg 2\Delta/R^2$ (large enough) in Algorithm 2, we obtain Figure 1. In this plot, we observe the distinctive accelerated convergence rate of Algorithm 2 with non-monotonic behavior, supporting our theoretical results.

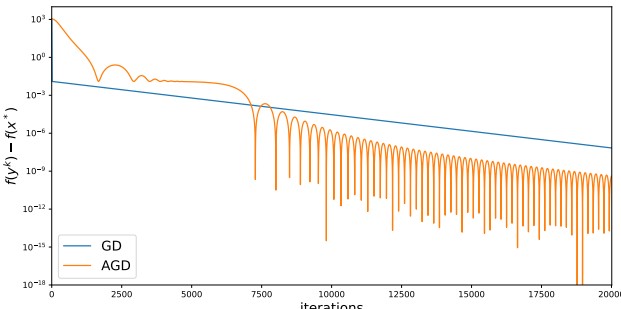

Figure 1: Experiment with $e^x + e^{1-x} + \frac{\mu}{2} y^2$ and $\mu = 0.001$

## A.2  COMPARISON WITH PREVIOUS AGD METHODS

Using the same function and setup, we compare our Algorithm 2 with previous accelerated methods in Figure 2. For all methods, we choose parameter values according to the theorems in their respective papers. Notice that AGD by Vankov et al. (2024) requires a method that solves an auxiliary problem. To solve this problem, we use binary search with 10 and 100 steps. In Figure 2, we observe very different behaviors across the methods. AGD by Li et al. (2024a) has the slowest convergence since their method chooses a small step size. The method by Vankov et al. (2024) is very sensitive to the number of inner steps used to solve the auxiliary problem: with only inner step 10 steps, it converges slowly. At the beginning, the method by Gorbunov et al. (2025) has the fastest convergence, while our method performs better at lower accuracies.

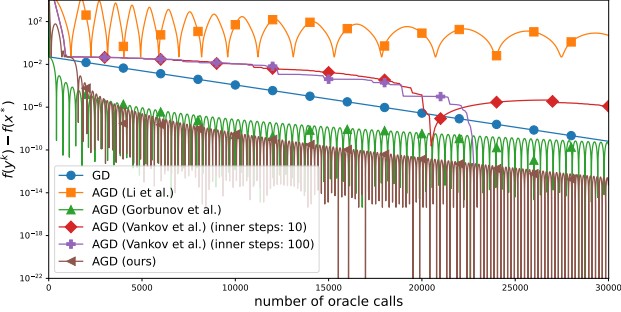

Figure 2: Experiment with $e^x + e^{1-x} + \frac{\mu}{2} y^2$ and $\mu = 0.001$

## A.3  SENSITIVITY TO THE CHOICE OF $\bar{R}$ AND $\Gamma_0$

We now also check how sensitive our algorithm is to the choice of $\bar{R}$ and $\Gamma_0$. In Figures 3 and 4, we fix the theoretically best values and increase them by $5\times$ and $25\times$. We observe that the algorithm is not very sensitive to the choice of $\Gamma_0$, but more sensitive to the choice of $\bar{R}$, which is expected since $\Gamma_0$ is under the logarithms in (13), while $\bar{R}$ is not.

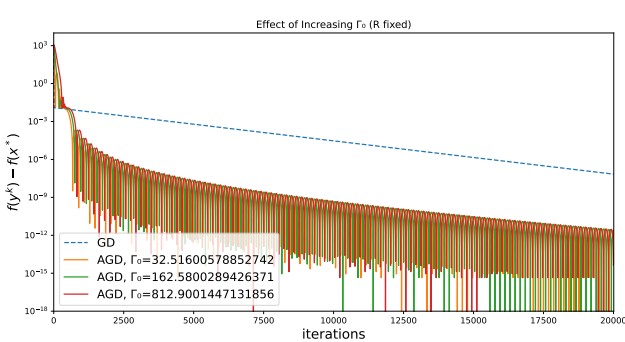

Figure 3: Sensitivity to increasing $\Gamma_0$ by $5\times$ and $25\times$.

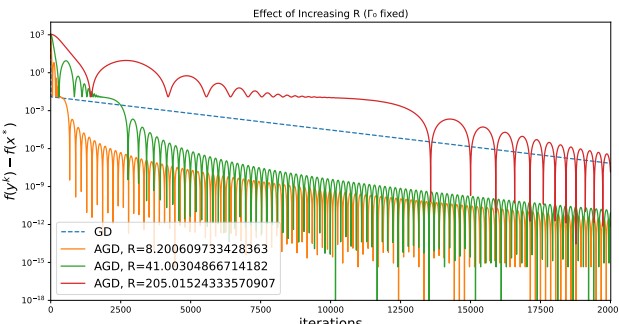

Figure 4: Sensitivity to increasing $\bar{R}$ by $5\times$ and $25\times$.

## A.4 EXPERIMENTS WITH ALGORITHM 1 AND NON-MONOTONIC $\psi$

We now consider Algorithm 1 and the results from Section 5. We take the function $f : \mathbb{R}^2 \to \mathbb{R}$ defined as $f(x, y) = -\sqrt{x} - \sqrt{1-x} + \frac{\mu}{2}y^2$, where $\mu = 0.001$, which is $(3, 4, 10)$–smooth. For this function, we can only use Algorithm 1 with the corresponding non-monotonic $\psi$. We start at $x^0 = (0.3, -0.15)$ and take $\bar{R} = R$ in Algorithm 1. Unlike Algorithm 2, we have to choose $\delta$. We can take $M_{\bar{R}} = 4.47 \geq \max_{\|x-x^*\| \leq 2\bar{R}} \|\nabla f(x)\|$, which we estimated numerically. Then, we choose $\delta$ according to (52), where the latter choice was derived for $(\rho, L_0, L_1)$–smooth functions. The results are presented in Figure 5. In practice, we observe that the required number of GD steps is small, less than 10, and thus the GD iterations in Algorithm 1 are almost invisible in the plot. Similarly to Section A.1, AGD converges non-monotonically faster than GD.

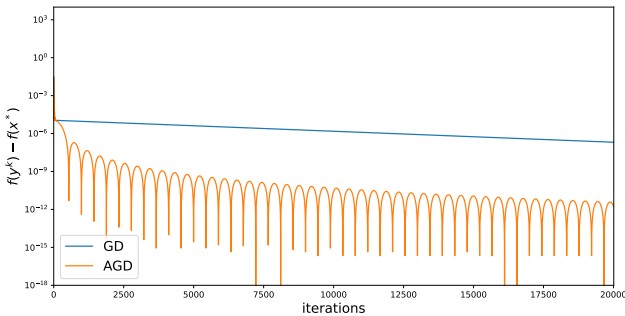

Figure 5: Experiment with $-\sqrt{x} - \sqrt{1-x} + \frac{\mu}{2}y^2$ and $\mu = 0.001$

# B   AUXILIARY LEMMAS

In the proofs, we use the following useful lemma from (Tyurin, 2025), which generalizes the key inequality from Theorem 2.1.5 of (Nesterov, 2018).

**Lemma B.1** (Tyurin (2025)). *For all $x, y \in \mathcal{X}$, if $f$ is $\ell$–smooth (Assumption 2.1) and convex (Assumption 2.3), then*

$$\|\nabla f(x) - \nabla f(y)\|^2 \int_0^1 \frac{1-v}{\ell(\|\nabla f(x)\| + \|\nabla f(x) - \nabla f(y)\| v)} dv \le f(x) - f(y) - \langle \nabla f(y), x - y \rangle. \tag{15}$$

The following lemma ensures that it is "safe" to take steps with proper step sizes.

**Lemma B.2** (Tyurin (2025)). *Under Assumption 2.1, for a fixed $x \in \mathcal{X}$, the point $y = x + th \in \mathcal{X}$ for all $t \in \left[0, \int_0^\infty \frac{dv}{\ell(\|\nabla f(x)\| + v)}\right)$ and $h \in \mathbb{R}^d$ such that $\|h\| = 1$.*

We now prove two important lemmas that allow us to bound the norm $\|\nabla f(y)\|$ given an upper bound on $f(y) - f(x^*)$.

**Lemma B.3.** *[Strictly Increasing $\psi$] Under Assumptions 2.1 and 2.3, let $f(y) - f(x^*) \le \delta$ for some $y \in \mathcal{X}, \delta > 0$ and $\psi : \mathbb{R}_+ \to \mathbb{R}_+$ such that $\psi(x) = \frac{x^2}{2\ell(4x)}$ is strictly increasing, then $\|\nabla f(y)\| \le \psi^{-1}(\delta)$ if $\delta \in \text{im}(\psi)$.*

*Proof.* Using Lemma B.1 and the fact that $\ell$ is non-decreasing,

$$\delta \ge f(y) - f(x^*) \ge \|\nabla f(y)\|^2 \int_0^1 \frac{1-v}{\ell(\|\nabla f(y)\| + \|\nabla f(y)\| v)} dv$$

$$\ge \frac{\|\nabla f(y)\|^2}{2\ell(4\|\nabla f(y)\|)} = \psi(\|\nabla f(y)\|).$$

It left to invert $\psi$ to get the result.  □

**Lemma B.4.** *[Not Necessarily Strictly Increasing $\psi$] Under Assumptions 2.1 and 2.3, let $\psi : \mathbb{R}_+ \to \mathbb{R}_+$ such that $\psi(x) = \frac{x^2}{2\ell(4x)}$ is not necessarily strictly increasing.*

1. *There exists the largest $\Delta_{\max} \in (0, \infty]$ such that $\psi$ is strictly increasing on $[0, \Delta_{\max})$,*

2. *For all $\delta \in [0, \psi(\Delta_{\max}))$, there exists the unique $\Delta_{\text{left}}(\delta) \in [0, \Delta_{\max})$ and the smallest[5] $\Delta_{\text{right}}(\delta) \in [\Delta_{\max}, \infty]$ such that $\psi(\Delta_{\text{left}}(\delta)) = \delta$ and $\psi(\Delta_{\text{right}}(\delta)) = \delta$.*

3. *For all $\delta \in [0, \psi(\Delta_{\max}))$, if $\Delta_{\text{right}}(\delta) < \infty$ and $\delta > \bar{\delta} \ge 0$, then $\Delta_{\text{right}}(\bar{\delta}) > \Delta_{\text{right}}(\delta)$.*

4. *If $f(y) - f(x^*) \le \delta$ for some $y \in \mathcal{X}$ and $\delta \in [0, \psi(\Delta_{\max}))$, then either $\|\nabla f(y)\| \le \Delta_{\text{left}}(\delta)$ or $\|\nabla f(y)\| \ge \Delta_{\text{right}}(\delta)$.*

*Proof.* 1. Since $\ell$ is non-decreasing and locally Lipschitz, there exists $\bar{\Delta}_1 > 0$ such that

$$2\ell(4y) - 2\ell(4x) \le M(y - x)$$

for all $0 \le x < y \le \bar{\Delta}_1$ and for some $M \equiv M(\bar{\Delta}_1, \ell) > 0$. Thus,

$$x^2 2\ell(4y) \le x^2 2\ell(4x) + Mx^2(y - x). \tag{16}$$

Moreover, there exists $\bar{\Delta}_2 > 0$ such that

$$Mx^2 < (y + x)2\ell(4x)$$

---

[5]if the set $\{x \in [\Delta_{\max}, \infty) : \psi(x) = \delta\}$ is empty, then $\Delta_{\text{right}}(\delta) = \infty$

for all $0 \leq x < y \leq \bar{\Delta}_2$ since $2\ell(4x) \geq \ell(0) > 0$, the l.h.s $\mathcal{O}(x^2)$, and the r.h.s. $\Omega(x)$. Combining with (16),

$$x^2 2\ell(4y) < x^2 2\ell(4x) + 2\ell(4x)(y+x)(y-x) = y^2 2\ell(4x)$$

and

$$\frac{x^2}{2\ell(4x)} < \frac{y^2}{2\ell(4y)}$$

for all $0 \leq x < y \leq \min\{\bar{\Delta}_1, \bar{\Delta}_2\}$, meaning that $\psi$ is locally strictly increasing on the interval $[0, \Delta_{\max})$ for some largest $\Delta_{\max} \in (0, \infty]$.

2. $\Delta_{\text{left}}(\delta)$ exists since $\psi$ is locally strictly increasing on the interval $[0, \Delta_{\max})$. On the interval $[\Delta_{\max}, \infty)$, either $\psi$ intersects $\delta$ for the first time at $\Delta_{\text{right}}(\delta)$ or we can take $\Delta_{\text{right}}(\delta) = \infty$.

3. Since $\Delta_{\text{right}}(\delta)$ is the first time when $\psi$ intersects $\delta$ for $x \in [\Delta_{\max}, \infty)$ and $\delta < \psi(\Delta_{\max})$, then $\psi(x) > \delta$ for all $x \in [\Delta_{\max}, \Delta_{\text{right}}(\delta))$. Thus, if we decrease $\delta$ and take $\bar{\delta} < \delta$, then $\Delta_{\text{right}}(\bar{\delta})$ can only increase or stay the same. However, if $\Delta_{\text{right}}(\bar{\delta})$ stays the same, i.e., $\Delta_{\text{right}}(\bar{\delta}) = \Delta_{\text{right}}(\delta)$, then $\Delta_{\text{right}}(\bar{\delta})$ is the first time when $\psi$ intersects $\delta$, which is impossible due to the continuity of $\psi$ and the fact that $\Delta_{\text{right}}(\bar{\delta})$ is the first time when $\psi$ intersects $\bar{\delta} < \delta$.

4. Using the same reasoning as in the proof of Lemma B.3:

$$\delta \geq \psi\left(\|\nabla f(y)\|\right). \tag{17}$$

Due to the previous properties, either $\|\nabla f(y)\| \leq \Delta_{\text{left}}(\delta)$ or $\|\nabla f(y)\| \geq \Delta_{\text{right}}(\delta)$ because $\psi(x) > \delta$ for all $x \in (\Delta_{\text{left}}(\delta), \Delta_{\text{right}}(\delta))$. $\qquad\square$

## C  RATE OF THE AUXILIARY SEQUENCE

**Theorem 3.1.** *For any $\Gamma_0 > 0$ and $\gamma \geq 0$, let $\alpha_k \geq \sqrt{\gamma \Gamma_k}$ and $\Gamma_{k+1} = \Gamma_k/(1 + \alpha_k)$ for all $k \geq 0$. Then, $\Gamma_{k+1} \leq \frac{9}{\gamma\left(k+1-\bar{k}\right)^2}$ for all $k \geq \bar{k} := \max\left\{1 + \frac{1}{2}\log_{3/2}\left(\frac{\gamma \Gamma_0}{4}\right), 0\right\}$.*

*Proof.* By the definition of $\Gamma_{k+1}$ and $\alpha_k$,

$$\Gamma_{k+1} \leq \frac{\Gamma_k}{1 + \sqrt{\gamma \Gamma_k}}$$

for all $k \geq 0$. Instead of $\Gamma_k$, consider the sequence $\bar{\Gamma}_k$ such that

$$\bar{\Gamma}_{k+1} = \frac{\bar{\Gamma}_k}{1 + \sqrt{\gamma \bar{\Gamma}_k}}$$

for all $k \geq 0$ and $\bar{\Gamma}_0 = \Gamma_0$. Using mathematical induction, notice that $\bar{\Gamma}_{k+1} \geq \Gamma_{k+1}$. Indeed, the function $\frac{x}{1 + \sqrt{\gamma x}}$ is increasing[6] for all $x \geq 0$ and

$$\Gamma_{k+1} \leq \frac{\Gamma_k}{1 + \sqrt{\gamma \Gamma_k}} \leq \frac{\bar{\Gamma}_k}{1 + \sqrt{\gamma \bar{\Gamma}_k}} = \bar{\Gamma}_{k+1}$$

if $\Gamma_k \leq \bar{\Gamma}_k$. If we bound $\bar{\Gamma}_{k+1}$, then we can bound $\Gamma_{k+1}$. Next,

$$\frac{1}{\bar{\Gamma}_{k+1}} - \frac{1}{\bar{\Gamma}_k} = \sqrt{\frac{\gamma}{\bar{\Gamma}_k}}$$

Let us define $t_k := \frac{1}{\bar{\Gamma}_k}$ for all $k \geq 0$, then

$$t_{k+1} - t_k = \sqrt{\gamma t_k}. \tag{18}$$

---

[6]$\left(\frac{x}{1+\sqrt{\gamma x}}\right)' = \frac{1 + \frac{\sqrt{\gamma x}}{2}}{(1+\sqrt{\gamma x})^2} > 0$ for all $x \geq 0$.

and

$$(t_{k+1}^{1/2} + t_k^{1/2})(t_{k+1}^{1/2} - t_k^{1/2}) = \sqrt{\gamma t_k} \tag{19}$$

for all $k \geq 0$. We now fix any $k \geq 0$. There are two options:

**Option 1:** $t_k^{1/2} \leq \frac{\sqrt{\gamma}}{2}$.

In this case, using (18),

$$t_{k+1} = t_k + \sqrt{\gamma t_k} \leq \frac{\gamma}{4} + \frac{\gamma}{2} = \frac{3\gamma}{4}$$

and

$$2\sqrt{\gamma}(t_{k+1}^{1/2} - t_k^{1/2}) \geq \sqrt{\gamma t_k}$$

due to (19). Rearranging the terms,

$$t_{k+1}^{1/2} \geq \frac{3}{2} t_k^{1/2} \geq \left(\frac{3}{2}\right)^{k+1} t_0^{1/2}, \tag{20}$$

where we unroll the recursion since $t_0^{1/2} \leq \cdots \leq t_k^{1/2} \leq \frac{\sqrt{\gamma}}{2}$.

**Option 2:** $t_k^{1/2} > \frac{\sqrt{\gamma}}{2}$.

Using (18),

$$t_{k+1} = t_k + \sqrt{\gamma t_k} \leq t_k + 2t_k \leq 3t_k$$

and

$$3t_k^{1/2}(t_{k+1}^{1/2} - t_k^{1/2}) \geq \sqrt{\gamma t_k}$$

due to (19), which yields

$$t_{k+1}^{1/2} \geq t_k^{1/2} + \frac{\sqrt{\gamma}}{3}. \tag{21}$$

Let $k^* \geq 0$ be the smallest index such that $t_{k^*}^{1/2} > \frac{\sqrt{\gamma}}{2}$. Unrolling (21),

$$t_{k+1}^{1/2} \geq t_{k^*}^{1/2} + (k + 1 - k^*)\frac{\sqrt{\gamma}}{3} \tag{22}$$

for all $k \geq k^*$. If $k^* = 0$, then

$$t_{k+1}^{1/2} \geq (k + 1)\frac{\sqrt{\gamma}}{3}. \tag{23}$$

Otherwise, by the definition of $k^*$,

$$\left(\frac{3}{2}\right)^{k^*-1} t_0^{1/2} \overset{(20)}{\leq} t_{k^*-1}^{1/2} \leq \frac{\sqrt{\gamma}}{2},$$

which yields

$$k^* \leq 1 + \frac{1}{2}\log_{3/2}\left(\frac{\gamma}{4t_0}\right)$$

and

$$t_{k+1}^{1/2} \geq \left(k + 1 - \left(1 + \frac{1}{2}\log_{3/2}\left(\frac{\gamma}{4t_0}\right)\right)\right)\frac{\sqrt{\gamma}}{3}, \tag{24}$$

due to (22). Combining the cases with $k^* = 0$ and $k^* > 0$, we get

$$t_{k+1}^{1/2} \geq \left(k + 1 - \bar{k}\right)\frac{\sqrt{\gamma}}{3} \tag{25}$$

for all $k \geq \bar{k} := \max\left\{1 + \frac{1}{2}\log_{3/2}\left(\frac{\gamma}{4t_0}\right), 0\right\}$. It left to recall that $t_k = 1/\bar{\Gamma}_k$ and $\bar{\Gamma}_k \geq \Gamma_k$ for all $k \geq 0$ to obtain the result.

$\square$

## D    MAIN DESCENT LEMMA

**Lemma D.1.** *Suppose that Assumptions 2.1 and 2.3 hold. Consider Algorithm 1 up to the $k^{th}$ iteration and the following virtual steps:*

$$\alpha_k(\gamma) \equiv \alpha_{k,\gamma} = \sqrt{\gamma \Gamma_k},$$

$$y^{k+1}(\gamma) \equiv y_\gamma^{k+1} = \frac{1}{1+\alpha_{k,\gamma}} y^k + \frac{\alpha_{k,\gamma}}{1+\alpha_{k,\gamma}} u^k - \frac{\gamma}{1+\alpha_{k,\gamma}} \nabla f(y^k),$$

$$u^{k+1}(\gamma) \equiv u_\gamma^{k+1} = \text{Proj}_{\bar{\mathcal{X}}}\left( u^k - \frac{\alpha_{k,\gamma}}{\Gamma_k} \nabla f(y_\gamma^{k+1}) \right), \tag{26}$$

$$\Gamma_{k+1}(\gamma) \equiv \Gamma_{k+1,\gamma} = \Gamma_k/(1+\alpha_{k,\gamma}),$$

*where $0 \le \gamma \le \frac{1}{\ell(2\|\nabla f(y^k)\|)}$ is a free parameter, $y^k \in \mathcal{X}$, and $u^k \in \bar{\mathcal{X}}$. Then, the steps (26) are well-defined, $y_\gamma^{k+1} \in \mathcal{X}$, and $u_\gamma^{k+1} \in \bar{\mathcal{X}}$, and*

$$(1+\alpha_{k,\gamma})(f(y_\gamma^{k+1}) - f(x^*)) + \frac{(1+\alpha_{k,\gamma})\Gamma_{k+1,\gamma}}{2} \left\| u_\gamma^{k+1} - x^* \right\|^2 - \left( (f(y^k) - f(x^*)) + \frac{\Gamma_k}{2} \left\| u^k - x^* \right\|^2 \right)$$

$$\le \frac{1}{2}\left( \gamma - \frac{1}{\ell(2\|\nabla f(y^k)\| + \|\nabla f(y_\gamma^{k+1})\|)} \right) \left\| \nabla f(y_\gamma^{k+1}) - \nabla f(y^k) \right\|^2.$$

*Proof.* (The following steps up to (27) may be skipped by the reader if $\mathcal{X} = \mathbb{R}^n$)
Clearly, $u_\gamma^{k+1} \in \bar{\mathcal{X}}$ due the projection operator. However, we have to check that $y_\gamma^{k+1} \in \mathcal{X}$ to make sure the steps are well-defined. Notice that

$$y_\gamma^{k+1} = \frac{1}{1+\alpha_{k,\gamma}}\left( y^k - \gamma \nabla f(y^k) \right) + \frac{\alpha_{k,\gamma}}{1+\alpha_{k,\gamma}} u^k$$

Moreover, $y^k - \gamma \nabla f(y^k) \in \mathcal{X}$. If $\nabla f(y^k) = 0$, then it is trivial. Otherwise,

$$y^k - \gamma \nabla f(y^k) = y^k - \gamma \left\| \nabla f(y^k) \right\| \frac{\nabla f(y^k)}{\|\nabla f(y^k)\|} \in \mathcal{X}$$

due to Lemma B.2 because

$$\gamma \left\| \nabla f(y^k) \right\| \le \frac{\left\| \nabla f(y^k) \right\|}{\ell(2\|\nabla f(y^k)\|)} \le \int_0^\infty \frac{dv}{\ell(\|\nabla f(y^k)\| + v)}.$$

for all $\gamma \le \frac{1}{\ell(2\|\nabla f(y^k)\|)}$. In total, $y_\gamma^{k+1} \in \mathcal{X}$ since $\mathcal{X}$ is an open convex set, $u^k \in \bar{\mathcal{X}}$, and $\frac{1}{1+\alpha_{k,\gamma}} \ne 0$ (as it is a convex combination of a point from $\mathcal{X}$ and a point from $\bar{\mathcal{X}}$ with a non-zero weight; see (Rockafellar, 2015)[Theorem 6.1]).

Consider the difference

$$f(y_\gamma^{k+1}) - f(x^*) + \frac{\Gamma_{k+1,\gamma}}{2} \left\| u_\gamma^{k+1} - x^* \right\|^2 - \left( (f(y^k) - f(x^*)) + \frac{\Gamma_k}{2} \left\| u^k - x^* \right\|^2 \right). \tag{27}$$

Rearranging the terms, we get

$$f(y_\gamma^{k+1}) - f(x^*) + \frac{\Gamma_{k+1,\gamma}}{2} \left\| u_\gamma^{k+1} - x^* \right\|^2 - \left( (f(y^k) - f(x^*)) + \frac{\Gamma_k}{2} \left\| u^k - x^* \right\|^2 \right)$$

$$= -(f(y^k) - f(y_\gamma^{k+1}) - \left\langle \nabla f(y_\gamma^{k+1}), y^k - y_\gamma^{k+1} \right\rangle)$$

$$+ \left\langle \nabla f(y_\gamma^{k+1}), y_\gamma^{k+1} - y^k \right\rangle$$

$$+ \frac{\Gamma_{k+1,\gamma} - \Gamma_k}{2} \left\| u_\gamma^{k+1} - x^* \right\|^2 + \frac{\Gamma_k}{2}\left( \left\| u_\gamma^{k+1} - x^* \right\|^2 - \left\| u^k - x^* \right\|^2 \right).$$

Since $\Gamma_k = (1+\alpha_{k,\gamma})\Gamma_{k+1,\gamma}$,

$$f(y_\gamma^{k+1}) - f(x^*) + \frac{(1+\alpha_{k,\gamma})\Gamma_{k+1,\gamma}}{2} \left\| u_\gamma^{k+1} - x^* \right\|^2 - \left( (f(y^k) - f(x^*)) + \frac{\Gamma_k}{2} \left\| u^k - x^* \right\|^2 \right)$$

$$= -(f(y^k) - f(y_\gamma^{k+1})) - \langle \nabla f(y_\gamma^{k+1}), y^k - y_\gamma^{k+1} \rangle$$
$$+ \langle \nabla f(y_\gamma^{k+1}), y_\gamma^{k+1} - y^k \rangle$$
$$+ \frac{\Gamma_k}{2} \left( \left\| u_\gamma^{k+1} - x^* \right\|^2 - \left\| u^k - x^* \right\|^2 \right).$$

Due to $\|a\|^2 - \|a + b\|^2 = -\|b\|^2 - 2\langle a, b\rangle$ for all $a, b \in \mathbb{R}^d$,

$$f(y_\gamma^{k+1}) - f(x^*) + \frac{(1 + \alpha_{k,\gamma})\Gamma_{k+1,\gamma}}{2} \left\| u_\gamma^{k+1} - x^* \right\|^2 - \left( (f(y^k) - f(x^*)) + \frac{\Gamma_k}{2} \left\| u^k - x^* \right\|^2 \right)$$
$$= -(f(y^k) - f(y_\gamma^{k+1})) - \langle \nabla f(y_\gamma^{k+1}), y^k - y_\gamma^{k+1} \rangle$$
$$+ \langle \nabla f(y_\gamma^{k+1}), y_\gamma^{k+1} - y^k \rangle$$
$$+ \frac{\Gamma_k}{2} \left( - \left\| u^k - u_\gamma^{k+1} \right\|^2 - 2 \langle u_\gamma^{k+1} - x^*, u^k - u_\gamma^{k+1} \rangle \right).$$

$$(28)$$

Consider the last inner product:

$$- \langle u_\gamma^{k+1} - x^*, u^k - u_\gamma^{k+1} \rangle$$
$$= \left\langle u_\gamma^{k+1} - x^*, \left( u^k - \frac{\alpha_{k,\gamma}}{\Gamma_k} \nabla f(y_\gamma^{k+1}) \right) - u^k \right\rangle + \left\langle u_\gamma^{k+1} - x^*, u_\gamma^{k+1} - \left( u^k - \frac{\alpha_{k,\gamma}}{\Gamma_k} \nabla f(y_\gamma^{k+1}) \right) \right\rangle.$$

Using $u_\gamma^{k+1} = \mathrm{Proj}_{\bar{\mathcal{X}}} \left( u^k - \frac{\alpha_{k,\gamma}}{\Gamma_k} \nabla f(y_\gamma^{k+1}) \right)$ and the projection property $\langle \mathrm{Proj}_{\bar{\mathcal{X}}}(y) - x, \mathrm{Proj}_{\bar{\mathcal{X}}}(y) - y \rangle \le 0$ for all $y \in \mathbb{R}^d, x \in \bar{\mathcal{X}}$, we have

$$- \langle u_\gamma^{k+1} - x^*, u^k - u_\gamma^{k+1} \rangle \le \left\langle u_\gamma^{k+1} - x^*, \left( u^k - \frac{\alpha_{k,\gamma}}{\Gamma_k} \nabla f(y_\gamma^{k+1}) \right) - u^k \right\rangle = - \left\langle u_\gamma^{k+1} - x^*, \frac{\alpha_{k,\gamma}}{\Gamma_k} \nabla f(y_\gamma^{k+1}) \right\rangle.$$

Substituting to (28),

$$f(y_\gamma^{k+1}) - f(x^*) + \frac{(1 + \alpha_{k,\gamma})\Gamma_{k+1,\gamma}}{2} \left\| u_\gamma^{k+1} - x^* \right\|^2 - \left( (f(y^k) - f(x^*)) + \frac{\Gamma_k}{2} \left\| u^k - x^* \right\|^2 \right)$$
$$= -(f(y^k) - f(y_\gamma^{k+1})) - \langle \nabla f(y_\gamma^{k+1}), y^k - y_\gamma^{k+1} \rangle$$
$$+ \langle \nabla f(y_\gamma^{k+1}), y_\gamma^{k+1} - y^k \rangle$$
$$+ \frac{\Gamma_k}{2} \left( - \left\| u^k - u_\gamma^{k+1} \right\|^2 - 2 \left\langle u_\gamma^{k+1} - x^*, \frac{\alpha_{k,\gamma}}{\Gamma_k} \nabla f(y_\gamma^{k+1}) \right\rangle \right)$$
$$= -(f(y^k) - f(y_\gamma^{k+1})) - \langle \nabla f(y_\gamma^{k+1}), y^k - y_\gamma^{k+1} \rangle$$
$$+ \langle \nabla f(y_\gamma^{k+1}), y_\gamma^{k+1} - y^k \rangle$$
$$- \frac{\Gamma_k}{2} \left\| u^k - u_\gamma^{k+1} \right\|^2$$
$$- \alpha_{k,\gamma} \langle u_\gamma^{k+1} - x^*, \nabla f(y_\gamma^{k+1}) \rangle$$
$$= -(f(y^k) - f(y_\gamma^{k+1})) - \langle \nabla f(y_\gamma^{k+1}), y^k - y_\gamma^{k+1} \rangle$$
$$+ \langle \nabla f(y_\gamma^{k+1}), y_\gamma^{k+1} - y^k - \alpha_{k,\gamma}(u_\gamma^{k+1} - y_\gamma^{k+1}) \rangle$$
$$- \frac{\Gamma_k}{2} \left\| u^k - u_\gamma^{k+1} \right\|^2$$
$$- \alpha_{k,\gamma} \langle y_\gamma^{k+1} - x^*, \nabla f(y_\gamma^{k+1}) \rangle.$$

In the last two equalities, we rearranged terms. Using the convexity of $f$, we have $-(f(y_\gamma^{k+1}) - f(x^*)) \ge - \langle \nabla f(y_\gamma^{k+1}), y_\gamma^{k+1} - x^* \rangle$ and

$$(1 + \alpha_{k,\gamma})(f(y_\gamma^{k+1}) - f(x^*)) + \frac{(1 + \alpha_{k,\gamma})\Gamma_{k+1,\gamma}}{2} \left\| u_\gamma^{k+1} - x^* \right\|^2 - \left( (f(y^k) - f(x^*)) + \frac{\Gamma_k}{2} \left\| u^k - x^* \right\|^2 \right)$$
$$\le -(f(y^k) - f(y_\gamma^{k+1})) - \langle \nabla f(y_\gamma^{k+1}), y^k - y_\gamma^{k+1} \rangle$$
$$+ \langle \nabla f(y_\gamma^{k+1}), y_\gamma^{k+1} - y^k - \alpha_{k,\gamma}(u_\gamma^{k+1} - y_\gamma^{k+1}) \rangle$$

$$-\frac{\Gamma_k}{2}\left\|u^k - u_\gamma^{k+1}\right\|^2$$

$$= -(f(y^k) - f(y_\gamma^{k+1}) - \langle\nabla f(y_\gamma^{k+1}), y^k - y_\gamma^{k+1}\rangle)$$

$$+ \langle\nabla f(y_\gamma^{k+1}), (1 + \alpha_{k,\gamma})y_\gamma^{k+1} - y^k - \alpha_{k,\gamma}u_\gamma^{k+1}\rangle$$

$$- \frac{\Gamma_k}{2}\left\|u^k - u_\gamma^{k+1}\right\|^2.$$

In the last equality, we rearranged terms. Recall that

$$(1 + \alpha_{k,\gamma})y_\gamma^{k+1} - y^k = \alpha_{k,\gamma}u^k - \gamma\nabla f(y^k).$$

Thus,

$$(1 + \alpha_{k,\gamma})(f(y_\gamma^{k+1}) - f(x^*)) + \frac{(1 + \alpha_{k,\gamma})\Gamma_{k+1,\gamma}}{2}\left\|u_\gamma^{k+1} - x^*\right\|^2 - \left((f(y^k) - f(x^*)) + \frac{\Gamma_k}{2}\left\|u^k - x^*\right\|^2\right)$$

$$\leq -(f(y^k) - f(y_\gamma^{k+1}) - \langle\nabla f(y_\gamma^{k+1}), y^k - y_\gamma^{k+1}\rangle)$$

$$+ \alpha_{k,\gamma}\langle\nabla f(y_\gamma^{k+1}), u^k - u_\gamma^{k+1}\rangle - \gamma\langle\nabla f(y_\gamma^{k+1}), \nabla f(y^k)\rangle$$

$$- \frac{\Gamma_k}{2}\left\|u^k - u_\gamma^{k+1}\right\|^2$$

$$= -(f(y^k) - f(y_\gamma^{k+1}) - \langle\nabla f(y_\gamma^{k+1}), y^k - y_\gamma^{k+1}\rangle)$$

$$+ \alpha_{k,\gamma}\langle\nabla f(y_\gamma^{k+1}), u^k - u_\gamma^{k+1}\rangle$$

$$- \frac{\gamma}{2}\left\|\nabla f(y_\gamma^{k+1})\right\|^2 - \frac{\gamma}{2}\left\|\nabla f(y^k)\right\|^2 + \frac{\gamma}{2}\left\|\nabla f(y_\gamma^{k+1}) - \nabla f(y^k)\right\|^2$$

$$- \frac{\Gamma_k}{2}\left\|u^k - u_\gamma^{k+1}\right\|^2,$$

where we use $-\langle a, b\rangle = \frac{1}{2}\left\|a - b\right\|^2 - \frac{1}{2}\left\|a\right\|^2 - \frac{1}{2}\left\|b\right\|^2$ for all $a, b \in \mathbb{R}^d$. Using Young's inequality,

$$(1 + \alpha_{k,\gamma})(f(y_\gamma^{k+1}) - f(x^*)) + \frac{(1 + \alpha_{k,\gamma})\Gamma_{k+1,\gamma}}{2}\left\|u_\gamma^{k+1} - x^*\right\|^2 - \left((f(y^k) - f(x^*)) + \frac{\Gamma_k}{2}\left\|u^k - x^*\right\|^2\right)$$

$$\leq -(f(y^k) - f(y_\gamma^{k+1}) - \langle\nabla f(y_\gamma^{k+1}), y^k - y_\gamma^{k+1}\rangle)$$

$$+ \frac{\gamma}{2}\left\|\nabla f(y_\gamma^{k+1})\right\|^2 + \frac{\alpha_{k,\gamma}^2}{2\gamma}\left\|u^k - u_\gamma^{k+1}\right\|^2$$

$$- \frac{\gamma}{2}\left\|\nabla f(y_\gamma^{k+1})\right\|^2 - \frac{\gamma}{2}\left\|\nabla f(y^k)\right\|^2 + \frac{\gamma}{2}\left\|\nabla f(y_\gamma^{k+1}) - \nabla f(y^k)\right\|^2$$

$$- \frac{\Gamma_k}{2}\left\|u^k - u_\gamma^{k+1}\right\|^2$$

$$= -(f(y^k) - f(y_\gamma^{k+1}) - \langle\nabla f(y_\gamma^{k+1}), y^k - y_\gamma^{k+1}\rangle)$$

$$+ \frac{\alpha_{k,\gamma}^2}{2\gamma}\left\|u^k - u_\gamma^{k+1}\right\|^2 - \frac{\Gamma_k}{2}\left\|u^k - u_\gamma^{k+1}\right\|^2$$

$$- \frac{\gamma}{2}\left\|\nabla f(y^k)\right\|^2 + \frac{\gamma}{2}\left\|\nabla f(y_\gamma^{k+1}) - \nabla f(y^k)\right\|^2,$$

where the terms $\frac{\gamma}{2}\left\|\nabla f(y_\gamma^{k+1})\right\|^2$ are cancelled out. Since $\alpha_{k,\gamma} = \sqrt{\gamma\Gamma_k}$, the terms with $\left\|u^k - u_\gamma^{k+1}\right\|$ are also cancelled out and

$$(1 + \alpha_{k,\gamma})(f(y_\gamma^{k+1}) - f(x^*)) + \frac{(1 + \alpha_{k,\gamma})\Gamma_{k+1,\gamma}}{2}\left\|u_\gamma^{k+1} - x^*\right\|^2 - \left((f(y^k) - f(x^*)) + \frac{\Gamma_k}{2}\left\|u^k - x^*\right\|^2\right)$$

$$\leq -(f(y^k) - f(y_\gamma^{k+1}) - \langle\nabla f(y_\gamma^{k+1}), y^k - y_\gamma^{k+1}\rangle)$$

$$- \frac{\gamma}{2}\left\|\nabla f(y^k)\right\|^2 + \frac{\gamma}{2}\left\|\nabla f(y_\gamma^{k+1}) - \nabla f(y^k)\right\|^2$$

$$\leq -(f(y^k) - f(y_\gamma^{k+1}) - \langle\nabla f(y_\gamma^{k+1}), y^k - y_\gamma^{k+1}\rangle) + \frac{\gamma}{2}\left\|\nabla f(y_\gamma^{k+1}) - \nabla f(y^k)\right\|^2, \tag{29}$$

where the last inequality due to $\frac{\gamma}{2}\left\|\nabla f(y^k)\right\|^2 \geq 0$. Using Lemma B.1, we get

$$f(y^k) - f(y_\gamma^{k+1}) - \langle\nabla f(y_\gamma^{k+1}), y^k - y_\gamma^{k+1}\rangle$$

$$\geq \left\| \nabla f(y^k) - \nabla f(y_\gamma^{k+1}) \right\|^2 \int_0^1 \frac{1 - v}{\ell(\|\nabla f(y^k)\| + \|\nabla f(y^k) - \nabla f(y_\gamma^{k+1})\| v)} dv$$

$$\geq \left\| \nabla f(y^k) - \nabla f(y_\gamma^{k+1}) \right\|^2 \frac{1}{2\ell(\|\nabla f(y^k)\| + \|\nabla f(y^k) - \nabla f(y_\gamma^{k+1})\|)},$$

where we use that $\ell$ is non-decreasing and bounded the term in the denominator by the maximum possible value with $v = 1$. Using triangle's inequality,

$$f(y^k) - f(y_\gamma^{k+1}) - \langle \nabla f(y_\gamma^{k+1}), y^k - y_\gamma^{k+1} \rangle \geq \left\| \nabla f(y^k) - \nabla f(y_\gamma^{k+1}) \right\|^2 \frac{1}{2\ell(2\|\nabla f(y^k)\| + \|\nabla f(y_\gamma^{k+1})\|)},$$

Substituting to (29),

$$(1 + \alpha_{k,\gamma})(f(y_\gamma^{k+1}) - f(x^*)) + \frac{(1 + \alpha_{k,\gamma})\Gamma_{k+1,\gamma}}{2} \left\| u_\gamma^{k+1} - x^* \right\|^2 - \left( (f(y^k) - f(x^*)) + \frac{\Gamma_k}{2} \left\| u^k - x^* \right\|^2 \right)$$

$$\leq \frac{1}{2} \left( \gamma - \frac{1}{\ell(2\|\nabla f(y^k)\| + \|\nabla f(y_\gamma^{k+1})\|)} \right) \left\| \nabla f(y_\gamma^{k+1}) - \nabla f(y^k) \right\|^2.$$

$\square$

# E  CONVERGENCE THEOREMS

## E.1  SUBQUADRATIC AND QUADRATIC GROWTH OF $\ell$

**Lemma E.1.** *Under Assumptions 2.1 and 2.3, let $\psi : \mathbb{R}_+ \to \mathbb{R}_+$ such that $\psi(x) = \frac{x^2}{2\ell(4x)}$ is strictly increasing, $f(y) - f(x^*) \leq \delta$ for some $y \in \mathcal{X}$, and any $\delta \in (0, \infty]$ such that $\ell\left(8\sqrt{\delta\ell(0)}\right) \leq 2\ell(0)$, then $\ell(4\|\nabla f(y)\|) \leq 2\ell(0)$.*

*Proof.* With this choice of $\delta$, we get

$$\ell(4\|\nabla f(y)\|) \leq 2\ell(0)$$

because, due to $f(y) - f(x^*) \leq \delta$ and Lemma B.3,

$$\ell\left(4\|\nabla f(y^0)\|\right) \leq \ell\left(4\psi^{-1}(\delta)\right)$$

and

$$\ell\left(4\psi^{-1}(\delta)\right) \leq 2\ell(0) \Leftrightarrow \frac{\left(\psi^{-1}(\delta)\right)^2}{4\ell(0)} \leq \frac{\left(\psi^{-1}(\delta)\right)^2}{2\ell(4\psi^{-1}(\delta))} \overset{\psi(\psi^{-1}(\delta))=\delta}{\Leftrightarrow} \frac{\left(\psi^{-1}(\delta)\right)^2}{4\ell(0)} \leq \delta \Leftrightarrow \psi^{-1}(\delta) \leq 2\sqrt{\delta\ell(0)}$$

$$\Leftrightarrow \delta \leq \psi\left(2\sqrt{\delta\ell(0)}\right) \Leftrightarrow \delta \leq \frac{2\delta\ell(0)}{\ell\left(8\sqrt{\delta\ell(0)}\right)} \Leftrightarrow \ell\left(8\sqrt{\delta\ell(0)}\right) \leq 2\ell(0).$$

$$(30)$$

$\square$

**Theorem 3.2.** *Suppose that Assumptions 2.1 and 2.3 hold. Let $\psi : \mathbb{R}_+ \to \mathbb{R}_+$ such that $\psi(x) = \frac{x^2}{2\ell(4x)}$ be strictly increasing. Then Algorithm 1 guarantees that*

$$f(y^{k+1}) - f(x^*) \leq \Gamma_{k+1}\bar{R}^2 \leq \frac{18\ell(0)\bar{R}^2}{\left(k + 1 - \bar{k}\right)^2} \tag{4}$$

*for all $k \geq \bar{k} := \max\left\{1 + \frac{1}{2}\log_{3/2}\left(\frac{\Gamma_0}{8\ell(0)}\right), 0\right\}$ with any $\delta \in (0, \infty]$ such that $\ell\left(8\sqrt{\delta\ell(0)}\right) \leq 2\ell(0)$ and any $\bar{R} \geq R := \|x^0 - x^*\|$.*

*Proof.* In our proof, we define the Lyapunov function $V_k := f(y^k) - f(x^*) + \frac{\Gamma_k}{2} \left\| u^k - x^* \right\|^2$. After running GD, we get $\ell \left( 4 \left\| \nabla f(y^0) \right\| \right) \leq 2\ell(0)$ due to Lemma E.1 and the choice of $\delta$. Trivially, $V_0 \leq V_0$. Due to $f(y^0) - f(x^*) \leq \frac{\delta}{2}$ in Alg. 1 and $\left\| y^0 - x^* \right\| \leq \left\| x^0 - x^* \right\|$ (GD is monotonic; (Tyurin, 2025)[Lemma I.2]),

$$V_0 = f(y^0) - f(x^*) + \frac{\Gamma_0}{2} \left\| y^0 - x^* \right\|^2 \leq \frac{\delta}{2} + \frac{\Gamma_0}{2} \left\| y^0 - x^* \right\|^2$$

$$\leq \frac{\delta}{2} + \frac{\Gamma_0}{2} \left\| x^0 - x^* \right\|^2 \leq \delta \tag{31}$$

since $\Gamma_0 = \frac{\delta}{R^2}$ and $\bar{R} \geq \left\| x^0 - x^* \right\|$. Using mathematical induction, we assume that $\ell \left( 4 \left\| \nabla f(y^k) \right\| \right) \leq 2\ell(0)$ and $V_k \leq \left( \prod_{i=0}^{k-1} \frac{1}{1+\alpha_i} \right) V_0$ for some $k \geq 0$.

Consider Lemma D.1 and the steps (26). Then,

$$(1 + \alpha_{k,\gamma})(f(y_\gamma^{k+1}) - f(x^*)) + \frac{(1 + \alpha_{k,\gamma})\Gamma_{k+1,\gamma}}{2} \left\| u_\gamma^{k+1} - x^* \right\|^2 - \left( (f(y^k) - f(x^*)) + \frac{\Gamma_k}{2} \left\| u^k - x^* \right\|^2 \right)$$

$$\leq \frac{1}{2} \left( \gamma - \frac{1}{\ell(2 \left\| \nabla f(y^k) \right\| + \left\| \nabla f(y_\gamma^{k+1}) \right\|)} \right) \left\| \nabla f(y_\gamma^{k+1}) - \nabla f(y^k) \right\|^2,$$

where $0 \leq \gamma \leq \frac{1}{\ell(2\|\nabla f(y^k)\|)}$ is a free parameter. Let us take the smallest $\gamma$ such that

$$g(\gamma) := \gamma - \frac{1}{\ell(2 \left\| \nabla f(y^k) \right\| + \left\| \nabla f(y_\gamma^{k+1}) \right\|)} = 0$$

and denote is as $\gamma^*$. Such a choice exists because $g(\gamma)$ is continuous for all $\gamma \geq 0$ as a composition of continuous functions ($y_\gamma^{k+1}$ is a continuous function of $\gamma$), $g(0) = -\frac{1}{\ell(2\|\nabla f(y^k)\| + \|\nabla f(y_0^{k+1})\|)} < 0$, and

$$g(\bar{\gamma}) = \bar{\gamma} - \frac{1}{\ell(2 \left\| \nabla f(y^k) \right\| + \left\| \nabla f(y_{\bar{\gamma}}^{k+1}) \right\|)} \geq \bar{\gamma} - \frac{1}{\ell(2 \left\| \nabla f(y^k) \right\|)} = 0$$

for $\bar{\gamma} = \frac{1}{\ell(2\|\nabla f(y^k)\|)}$. Note that $\gamma^* \leq \frac{1}{\ell(2\|\nabla f(y^k)\|)}$. For all $\gamma \leq \gamma^*$, $g(\gamma) \leq 0$ and

$$(1 + \alpha_{k,\gamma})(f(y_\gamma^{k+1}) - f(x^*)) + \frac{(1 + \alpha_{k,\gamma})\Gamma_{k+1,\gamma}}{2} \left\| u_\gamma^{k+1} - x^* \right\|^2$$

$$\leq (f(y^k) - f(x^*)) + \frac{\Gamma_k}{2} \left\| u^k - x^* \right\|^2 =: V_k, \tag{32}$$

which ensures that

$$f(y_\gamma^{k+1}) - f(x^*) \leq V_k.$$

Recall that $V_k \leq V_0 \overset{(31)}{\leq} \delta$. It means that

$$f(y_\gamma^{k+1}) - f(x^*) \leq \delta$$

and

$$\ell \left( 4 \left\| \nabla f(y_\gamma^{k+1}) \right\| \right) \leq 2\ell(0)$$

for all $\gamma \leq \gamma^*$ due to Lemma E.1. Therefore, by the definition of $\gamma^*$ and using $\ell \left( 4 \left\| \nabla f(y^k) \right\| \right) \leq 2\ell(0)$,

$$\gamma^* = \frac{1}{\ell(2 \left\| \nabla f(y^k) \right\| + \left\| \nabla f(y_{\gamma^*}^{k+1}) \right\|)} \geq \frac{1}{\max\{\ell(4 \left\| \nabla f(y^k) \right\|), \ell(4 \left\| \nabla f(y_{\gamma^*}^{k+1}) \right\|)\}} \geq \frac{1}{2\ell(0)},$$

meaning that we can take $\gamma = \frac{1}{2\ell(0)}$ and (32) holds:

$$(1 + \alpha_{k,\gamma})(f(y_\gamma^{k+1}) - f(x^*)) + \frac{(1 + \alpha_{k,\gamma})\Gamma_{k+1,\gamma}}{2} \left\| u_\gamma^{k+1} - x^* \right\|^2 \leq V_k.$$

Notice that $\alpha_{k,\gamma} = \alpha_k$, $y_\gamma^{k+1} = y^{k+1}$, $\Gamma_{k+1,\gamma} = \Gamma_{k+1}$, and $u_\gamma^{k+1} = u^{k+1}$ with $\gamma = \frac{1}{2\ell(0)}$. Therefore,

$$(1 + \alpha_{k,\gamma})(f(y_\gamma^{k+1}) - f(x^*)) + \frac{(1 + \alpha_{k,\gamma})\Gamma_{k+1,\gamma}}{2} \left\| u_\gamma^{k+1} - x^* \right\|^2 = (1 + \alpha_k)V_{k+1},$$

$$\ell\left(4 \left\| \nabla f(y^{k+1}) \right\| \right) \le 2\ell(0),$$

and

$$V_{k+1} \le \frac{1}{1 + \alpha_k}V_k \le \left( \prod_{i=0}^{k} \frac{1}{1 + \alpha_i} \right) V_0,$$

We have proved the next step of the induction. Finally, for all $k \ge 0$,

$$f(y^{k+1}) - f(x^*) \le V_{k+1} \le \left( \prod_{i=0}^{k} \frac{1}{1 + \alpha_i} \right) \left( f(y^0) - f(x^*) + \frac{\Gamma_0}{2} \left\| y^0 - x^* \right\|^2 \right)$$

$$= \Gamma_0 \left( \prod_{i=0}^{k} \frac{1}{1 + \alpha_i} \right) \left( \frac{1}{\Gamma_0}(f(y^0) - f(x^*)) + \frac{1}{2} \left\| y^0 - x^* \right\|^2 \right).$$

Since $f(y^0) - f(x^*) \le \frac{\delta}{2}$, $\left\| y^0 - x^* \right\|^2 \le \left\| x^0 - x^* \right\|^2 \le \bar{R}^2$, and $\Gamma_{k+1} = \Gamma_0 \left( \prod_{i=0}^{k} \frac{1}{1+\alpha_i} \right)$,

$$f(y^{k+1}) - f(x^*) \le \Gamma_{k+1} \left( \frac{\delta}{2\Gamma_0} + \frac{1}{2}\bar{R}^2 \right) = \Gamma_{k+1}\bar{R}^2$$

because $\Gamma_0 = \frac{\delta}{\bar{R}^2}$. It is left to use Theorem 3.1. $\qquad \square$

**Theorem 3.3.** *Consider the assumptions and results of Theorem 3.2. The oracle complexity (i.e., the number of gradient calls) required to find an $\varepsilon$–solution is*

$$\frac{5\sqrt{\ell(0)}\bar{R}}{\sqrt{\varepsilon}} + k(\delta), \tag{5}$$

*for all $\delta \ge 0$ such that $\ell\left( 8\sqrt{\delta\ell(0)} \right) \le 2\ell(0)$, where $k(\delta) := \max\left\{ 1 + \frac{1}{2}\log_{3/2}\left( \frac{\delta}{8\ell(0)\bar{R}^2} \right), 0 \right\} + k_{\text{GD}}(\delta)$, $k_{\text{GD}}(\delta)$ is the oracle complexity of GD for finding a point $\bar{x}$ such that $f(\bar{x}) - f(x^*) \le \delta/2$.*

*Proof.* At the beginning, we run GD, which takes $k_{\text{GD}}(\delta)$ iterations (i.e., gradient evaluations). Next, using Theorem 3.1 and the choice of $\gamma = \frac{1}{2\ell(0)}$,

$$\Gamma_{k+1} \le \frac{18\ell(0)}{\left( k + 1 - \bar{k} \right)^2}$$

for all $k \ge \bar{k} := \max\left\{ 1 + \frac{1}{2}\log_{3/2}\left( \frac{\Gamma_0}{8\ell(0)} \right), 0 \right\}$. Taking

$$k \ge \frac{5\sqrt{\ell(0)}\bar{R}}{\sqrt{\varepsilon}} + \bar{k},$$

we get $f(y^{k+1}) - f(x^*) \le \varepsilon$ due to Theorem 3.2. $\qquad \square$

### E.2 STABILITY WITH RESPECT TO INPUT PARAMETERS AND IMPROVED RATES

**Theorem 4.1.** *Suppose that Assumptions 2.1 and 2.3 hold. Let $\psi : \mathbb{R}_+ \to \mathbb{R}_+$ such that $\psi(x) = \frac{x^2}{2\ell(4x)}$ be strictly increasing and $\lim_{x \to \infty} \psi(x) = \infty$. Then Algorithm 2 guarantees that*

$$f(y^{k+1}) - f(x^*) \le \Gamma_{k+1}R^2$$

*for all $k \ge 0$ with $\Gamma_0 \ge \frac{2(f(x^0) - f(x^*))}{\|x^0 - x^*\|^2}$ and $\bar{R} \ge R$.*

*Proof.* In our proof, we define the Lyapunov function $V_k := f(y^k) - f(x^*) + \frac{\Gamma_k}{2} \left\| u^k - x^* \right\|^2$. Trivially, $V_0 \leq V_0$ and

$$V_0 = f(y^0) - f(x^*) + \frac{\Gamma_0}{2} \left\| y^0 - x^* \right\|^2 \leq \Gamma_0 R^2 \leq \Gamma_0 \bar{R}^2 \tag{33}$$

when $\Gamma_0 \geq \frac{2(f(x^0) - f(x^*))}{\|x^0 - x^*\|^2} = \frac{2(f(y^0) - f(x^*))}{\|y^0 - x^*\|^2}$ and $\bar{R} \geq R$. Moreover,

$$f(y^0) - f(x^*) \leq \Gamma_0 \bar{R}^2.$$

Due to Lemma B.3,

$$\left\| \nabla f(y^0) \right\| \leq \psi^{-1}(\Gamma_0 \bar{R}^2).$$

Using mathematical induction, we assume that $\left\| \nabla f(y^k) \right\| \leq \psi^{-1}(\Gamma_k \bar{R}^2)$ and $V_k \leq \left( \prod_{i=0}^{k-1} \frac{1}{1+\alpha_i} \right) V_0$ for some $k \geq 0$.

Consider Lemma D.1 and the steps (26). Then,

$$(1 + \alpha_{k,\gamma})(f(y_\gamma^{k+1}) - f(x^*)) + \frac{(1 + \alpha_{k,\gamma})\Gamma_{k+1,\gamma}}{2} \left\| u_\gamma^{k+1} - x^* \right\|^2 - \left( (f(y^k) - f(x^*)) + \frac{\Gamma_k}{2} \left\| u^k - x^* \right\|^2 \right)$$

$$\leq \frac{1}{2} \left( \gamma - \frac{1}{\ell(2 \left\| \nabla f(y^k) \right\| + \left\| \nabla f(y_\gamma^{k+1}) \right\|)} \right) \left\| \nabla f(y_\gamma^{k+1}) - \nabla f(y^k) \right\|^2,$$

where $0 \leq \gamma \leq \frac{1}{\ell(2 \|\nabla f(y^k)\|)}$ is a free parameter. Let us take the smallest $\gamma$ such that

$$g(\gamma) := \gamma - \frac{1}{\ell(2 \left\| \nabla f(y^k) \right\| + \left\| \nabla f(y_\gamma^{k+1}) \right\|)} = 0$$

and denote is as $\gamma^*$ (exists similarly to the proof of Theorem 3.2 and $\gamma^* \leq \frac{1}{\ell(2\|\nabla f(y^k)\|)}$). For all $\gamma \leq \gamma^*$, $g(\gamma) \leq 0$ and

$$(1 + \alpha_{k,\gamma})(f(y_\gamma^{k+1}) - f(x^*)) + \frac{(1 + \alpha_{k,\gamma})\Gamma_{k+1,\gamma}}{2} \left\| u_\gamma^{k+1} - x^* \right\|^2$$

$$\leq (f(y^k) - f(x^*)) + \frac{\Gamma_k}{2} \left\| u^k - x^* \right\|^2 =: V_k. \tag{34}$$

Recall that

$$V_k \leq \left( \prod_{i=0}^{k-1} \frac{1}{1+\alpha_i} \right) V_0 = \frac{\Gamma_k}{\Gamma_0} V_0 \overset{(33)}{\leq} \Gamma_k \bar{R}^2.$$

Therefore,

$$f(y_\gamma^{k+1}) - f(x^*) \overset{(34)}{\leq} \frac{\Gamma_k \bar{R}^2}{1 + \alpha_{k,\gamma}}$$

and

$$\left\| \nabla f(y_\gamma^{k+1}) \right\| \leq \psi^{-1} \left( \frac{\Gamma_k \bar{R}^2}{1 + \alpha_{k,\gamma}} \right) \leq \psi^{-1} \left( \Gamma_k \bar{R}^2 \right) \tag{35}$$

for all $\gamma \leq \gamma^*$ due to Lemma B.3. Therefore, by the definition of $\gamma^*$ and using $\left\| \nabla f(y^k) \right\| \leq \psi^{-1}(\Gamma_k \bar{R}^2)$,

$$\gamma^* = \frac{1}{\ell(2 \left\| \nabla f(y^k) \right\| + \left\| \nabla f(y_{\gamma^*}^{k+1}) \right\|)} \geq \frac{1}{\max\{\ell(4 \left\| \nabla f(y^k) \right\|), \ell(4 \left\| \nabla f(y_{\gamma^*}^{k+1}) \right\|)\}} \geq \frac{1}{\ell\left(4\psi^{-1}\left(\Gamma_k \bar{R}^2\right)\right)},$$

meaning that we can take $\gamma_k = \frac{1}{\ell\left(4\psi^{-1}\left(\Gamma_k \bar{R}^2\right)\right)}$ and (32) holds:

$$(1 + \alpha_{k,\gamma})(f(y_\gamma^{k+1}) - f(x^*)) + \frac{(1 + \alpha_{k,\gamma})\Gamma_{k+1,\gamma}}{2} \left\| u_\gamma^{k+1} - x^* \right\|^2 \leq V_k.$$

Notice that $\alpha_{k,\gamma} = \alpha_k$, $y_\gamma^{k+1} = y^{k+1}$, $\Gamma_{k+1,\gamma} = \Gamma_{k+1}$, and $u_\gamma^{k+1} = u^{k+1}$ with $\gamma = \frac{1}{\ell\left(4\psi^{-1}\left(\Gamma_k\bar{R}^2\right)\right)}$.

Therefore, $(1 + \alpha_{k,\gamma})(f(y_\gamma^{k+1}) - f(x^*)) + \frac{(1+\alpha_{k,\gamma})\Gamma_{k+1,\gamma}}{2}\left\|u_\gamma^{k+1} - x^*\right\|^2 = (1 + \alpha_k)V_{k+1}$,

$$\left\|\nabla f(y^{k+1})\right\| \overset{(35)}{\leq} \psi^{-1}\left(\frac{\Gamma_k\bar{R}^2}{1 + \alpha_k}\right) = \psi^{-1}\left(\Gamma_{k+1}\bar{R}^2\right)$$

and

$$V_{k+1} \leq \frac{1}{1 + \alpha_k}V_k \leq \left(\prod_{i=0}^{k}\frac{1}{1 + \alpha_i}\right)V_0,$$

We have proved the next step of the induction. Finally, for all $k \geq 0$,

$$f(y^{k+1}) - f(x^*) \leq V_{k+1} \leq \Gamma_{k+1}\left(\frac{1}{\Gamma_0}(f(y^0) - f(x^*)) + \frac{1}{2}\left\|y^0 - x^*\right\|^2\right) \leq \Gamma_{k+1}\left\|x^0 - x^*\right\|^2$$

because $\Gamma_0 \geq \frac{2(f(y^0)-f(x^*))}{\|y^0-x^*\|^2}$, $\Gamma_{k+1} = \Gamma_0\left(\prod_{i=0}^{k}\frac{1}{1+\alpha_i}\right)$, and $y^0 = x^0$. $\qquad\square$

**Theorem 4.2.** *Consider the assumptions and results of Theorem 4.1. The oracle complexity (i.e., the number of gradient calls) required to find an $\varepsilon$–solution is*

$$\frac{5\sqrt{\ell(0)}R}{\sqrt{\varepsilon}} + \underbrace{\max\left\{2 + \log_{3/2}\left(\frac{\Gamma_0}{4\ell(0)}\right), 0\right\} + k_{\text{init}}}_{\text{does not depend on }\varepsilon} \tag{11}$$

*with $\Gamma_0 \geq \frac{2(f(x^0)-f(x^*))}{\|x^0-x^*\|^2}$, $\bar{R} \geq R$, and $k_{\text{init}}$ being the smallest integer such that*

$$\ell\left(24\sqrt{\frac{\ell\left(4\psi^{-1}\left(\Gamma_0\bar{R}^2\right)\right)\ell(0)\bar{R}^2}{k_{\text{init}}^2}}\right) \leq 2\ell(0).$$

*Proof.* Since $\gamma_k = 1/\ell\left(4\psi^{-1}\left(\Gamma_k\bar{R}^2\right)\right) \geq \gamma_0 := 1/\ell\left(4\psi^{-1}\left(\Gamma_0\bar{R}^2\right)\right)$ for all $k \geq 0$ in Algorithm 2, and by Theorem 3.1, we conclude that

$$\Gamma_k \leq \frac{9\ell\left(4\psi^{-1}\left(\Gamma_0\bar{R}^2\right)\right)}{\left(k - \bar{k}_1\right)^2} \tag{36}$$

for all $k > \bar{k}_1 := \max\left\{1 + \frac{1}{2}\log_{3/2}\left(\frac{\Gamma_0}{4\ell\left(4\psi^{-1}\left(\Gamma_0\bar{R}^2\right)\right)}\right), 0\right\}$. As in the proof of Lemma E.1 (take $\delta = \Gamma_k\bar{R}^2$ in (30)):

$$\ell\left(4\psi^{-1}\left(\Gamma_k\bar{R}^2\right)\right) \leq 2\ell(0) \Leftrightarrow \ell\left(8\sqrt{\Gamma_k\bar{R}^2\ell(0)}\right) \leq 2\ell(0). \tag{37}$$

Let $k_{\text{init}}$ be the smallest integer such that

$$\ell\left(24\sqrt{\frac{\ell\left(4\psi^{-1}\left(\Gamma_0\bar{R}^2\right)\right)\ell(0)\bar{R}^2}{k_{\text{init}}^2}}\right) \leq 2\ell(0).$$

Note that $k_{\text{init}} < \infty$, because $\ell$ is non-decreasing and continuous. Thus,

$$\ell\left(8\sqrt{\Gamma_k\bar{R}^2\ell(0)}\right) \leq 2\ell(0)$$

for all $k \geq k_{\text{init}} + \bar{k}_1$ due to (36), and $\gamma_k \geq \frac{1}{2\ell(0)}$ for all $k \geq k_{\text{init}} + \bar{k}_1$ due to (37). We now repeat the previous arguments once again. Using Theorem 3.1 with $\Gamma_0 \equiv \Gamma_{k_{\text{init}}+\bar{k}_1}$, we conclude that

$$\Gamma_{k+1+k_{\text{init}}+\bar{k}_1} \leq \frac{19\ell(0)}{\left(k + 1 - \bar{k}\right)^2}$$

for all $k \geq \bar{k} := \max\left\{1 + \frac{1}{2}\log_{3/2}\left(\frac{\Gamma_{k_{\text{init}}+\bar{k}_1}}{8\ell(0)}\right), 0\right\}$. It left to choose $k \geq \bar{k}$ such that

$$\frac{19\ell(0)R^2}{\left(k+1-\bar{k}\right)^2} \leq \varepsilon$$

and use Theorem 4.1 to get the total oracle complexity

$$\frac{5\sqrt{\ell(0)}R}{\sqrt{\varepsilon}} + \max\left\{1 + \frac{1}{2}\log_{3/2}\left(\frac{\Gamma_{k_{\text{init}}+\bar{k}_1}}{8\ell(0)}\right), 0\right\} + k_{\text{init}} + \max\left\{1 + \frac{1}{2}\log_{3/2}\left(\frac{\Gamma_0}{4\ell\left(4\psi^{-1}\left(\Gamma_0\bar{R}^2\right)\right)}\right), 0\right\}$$

$$\leq \frac{5\sqrt{\ell(0)}R}{\sqrt{\varepsilon}} + k_{\text{init}} + \max\left\{2 + \log_{3/2}\left(\frac{\Gamma_0}{4\ell(0)}\right), 0\right\}$$

because $\Gamma_k \leq \Gamma_0$ for all $k \geq 0$ and $\ell$ is non-decreasing. $\qquad\square$

### E.2.1 Specialization for $(L_0, L_1)$–smoothness

**Theorem 4.3.** *Consider the assumptions and results of Theorem 4.1 with $\ell(s) = L_0 + L_1 s$. The oracle complexity (i.e., the number of gradient calls) required to find an $\varepsilon$–solution is*

$$\mathcal{O}\left(\frac{\sqrt{L_0}R}{\sqrt{\varepsilon}} + \max\left\{L_1\bar{R}\log\left(\min\left\{\frac{L_1^2\bar{R}^2\Gamma_0}{L_0}, \frac{\Gamma_0 R^2}{\varepsilon}\right\}\right), 0\right\} + \max\left\{\log\left(\frac{\Gamma_0}{L_0}\right), 0\right\}\right) \quad (13)$$

*with $\Gamma_0 \geq \frac{2(f(x^0)-f(x^*))}{\|x^0-x^*\|^2}$ and $\bar{R} \geq R$.*

*Proof.* Since $\psi(x) = \frac{x^2}{2L_0+8L_1x}$, we get

$$\psi^{-1}(t) = 4L_1 t + \sqrt{16L_1^2 t^2 + 2L_0 t} \leq 8L_1 t + \sqrt{2L_0 t}$$

for all $t \geq 0$, and

$$\gamma_k = \frac{1}{\ell\left(4\psi^{-1}\left(\Gamma_k\bar{R}^2\right)\right)} \geq \frac{1}{L_0 + 4L_1(8L_1\Gamma_k\bar{R}^2 + \sqrt{2L_0\Gamma_k\bar{R}^2})}$$

$$= \frac{1}{L_0 + 32L_1^2\Gamma_k\bar{R}^2 + 4L_1\sqrt{2L_0\Gamma_k\bar{R}^2}} \overset{\text{AM-GM}}{\geq} \frac{1}{2L_0 + 48L_1^2\bar{R}^2\Gamma_k}.$$

Let $0 \leq k^* < \infty$ be the smallest $k$ such that $L_1^2\bar{R}^2\Gamma_k < L_0$. For all $k < k^*$, we get $L_1^2\bar{R}^2\Gamma_k \geq L_0$, $\gamma_k \geq \frac{1}{50L_1^2\bar{R}^2\Gamma_k}$, and $\alpha_k \geq \frac{1}{8L_1\bar{R}}$ since $\Gamma_k$ is decreasing. Then,

$$\Gamma_{k+1} \leq \frac{\Gamma_k}{1 + \frac{1}{8L_1\bar{R}}}.$$

for all $k < k^*$. We can unroll the recursion to get

$$\Gamma_{k+1} \leq \left(\frac{1}{1 + \frac{1}{8L_1\bar{R}}}\right)^{k+1}\Gamma_0 \leq \exp\left(-\frac{k+1}{8L_1\bar{R}+1}\right)\Gamma_0. \quad (38)$$

for all $k < k^*$. For all $k \geq k^*$, $L_1^2\bar{R}^2\Gamma_k < L_0$, $\gamma_k \geq \frac{1}{50L_0}$, and can we use Theorem 3.1 starting form the index $k^*$ :

$$\Gamma_{k+k^*} \leq \frac{450L_0}{\left(k-\bar{k}\right)^2}$$

for all $k > \bar{k}$, where

$$\bar{k} := \max\left\{1 + \frac{1}{2}\log_{3/2}\left(\frac{\Gamma_{k^*}}{200L_0}\right), 0\right\} \leq \max\left\{1 + \frac{1}{2}\log_{3/2}\left(\frac{\Gamma_0}{200L_0}\right), 0\right\}, \quad (39)$$

where the first inequality due to $\Gamma_{k^*} \leq \Gamma_0$. If $k^* = 0$, then

$$\Gamma_k \leq \frac{450 L_0}{\left(k - \bar{k}\right)^2}$$

for all $k > \bar{k}$. If $k^* > 0$, then

$$\frac{L_0}{L_1^2 \bar{R}^2} \leq \Gamma_{k^*-1} \overset{(38)}{\leq} \exp\left(-\frac{k^* - 1}{8 L_1 \bar{R} + 1}\right) \Gamma_0$$

and

$$k^* \leq 1 + \left(8 L_1 \bar{R} + 1\right) \log\left(\frac{L_1^2 \bar{R}^2 \Gamma_0}{L_0}\right).$$

In total,

$$k^* \leq \max\left\{1 + \left(8 L_1 \bar{R} + 1\right) \log\left(\frac{L_1^2 \bar{R}^2 \Gamma_0}{L_0}\right), 0\right\}. \tag{40}$$

There are two main regimes of $\Gamma_k$. The first regime is

$$\Gamma_k \leq \frac{450 L_0}{\left(k - \left(\bar{k} + k^*\right)\right)^2} \tag{41}$$

for all $k > \bar{k} + k^*$, and for all

$$k \geq \max\left\{1 + \left(8 L_1 \bar{R} + 1\right) \log\left(\frac{L_1^2 \bar{R}^2 \Gamma_0}{L_0}\right), 0\right\} + \max\left\{2 + 3 \log\left(\frac{\Gamma_0}{200 L_0}\right), 0\right\},$$

due to (39) and (40). The second regime is

$$\Gamma_k \leq \exp\left(-\frac{k}{8 L_1 \bar{R} + 1}\right) \Gamma_0 \tag{42}$$

for all $k \leq k^*$ due to (38).

Using Theorem 4.1,

$$f(y^{k+1}) - f(x^*) \leq \Gamma_{k+1} R^2.$$

If $\frac{L_1^2 \bar{R}^2 \Gamma_0}{L_0} \leq \frac{\Gamma_0 R^2}{\varepsilon}$, then $f(y^{k+1}) - f(x^*) \leq \varepsilon$ after

$$\mathcal{O}\left(\frac{\sqrt{L_0} R}{\sqrt{\varepsilon}} + \max\left\{\left(L_1 \bar{R} + 1\right) \log\left(\frac{L_1^2 \bar{R}^2 \Gamma_0}{L_0}\right), 0\right\} + \max\left\{\log\left(\frac{\Gamma_0}{L_0}\right), 0\right\}\right)$$

iterations due to (41). If $\frac{L_1^2 \bar{R}^2 \Gamma_0}{L_0} > \frac{\Gamma_0 R^2}{\varepsilon}$ and $k^* > \left(8 L_1 \bar{R} + 1\right) \log\left(\left(\Gamma_0 R^2\right)/\varepsilon\right)$, then $f(y^{k+1}) - f(x^*) \leq \varepsilon$ after

$$\mathcal{O}\left(\left(L_1 \bar{R} + 1\right) \log\left(\frac{\Gamma_0 R^2}{\varepsilon}\right)\right)$$

iterations due to (42). If $\frac{L_1^2 \bar{R}^2 \Gamma_0}{L_0} > \frac{\Gamma_0 R^2}{\varepsilon}$ and $k^* \leq \left(8 L_1 \bar{R} + 1\right) \log\left(\left(\Gamma_0 R^2\right)/\varepsilon\right)$, then $f(y^{k+1}) - f(x^*) \leq \varepsilon$ after

$$\mathcal{O}\left(\frac{\sqrt{L_0} R}{\sqrt{\varepsilon}} + \left(L_1 \bar{R} + 1\right) \log\left(\frac{\Gamma_0 R^2}{\varepsilon}\right) + \max\left\{\log\left(\frac{\Gamma_0}{L_0}\right), 0\right\}\right)$$

iterations due to (41). It left to combine all cases. $\qquad\square$

### E.3 SUPERQUADRATIC GROWTH OF $\ell$

**Theorem 5.1.** *Suppose that Assumptions 2.1 and 2.3 hold. Let $\psi : \mathbb{R}_+ \to \mathbb{R}_+$ such that $\psi(x) = \frac{x^2}{2\ell(4x)}$ be not necessarily strictly increasing. Find the largest $\Delta_{\max} \in (0, \infty]$ such that $\psi$ is strictly increasing on $[0, \Delta_{\max})$. For all $\delta \in [0, \psi(\Delta_{\max}))$, find the unique $\Delta_{\text{left}}(\delta) \in [0, \Delta_{\max})$ and the smallest[7] $\Delta_{\text{right}}(\delta) \in [\Delta_{\max}, \infty]$ such that $\psi(\Delta_{\text{left}}(\delta)) = \delta$ and $\psi(\Delta_{\text{right}}(\delta)) = \delta$. Take any $\delta \in [0, \frac{1}{2}\psi(\Delta_{\max})]$ such that $\ell(4\Delta_{\text{left}}(\delta)) \leq 2\ell(0)$ and $\Delta_{\text{right}}(\delta) \geq 2M_{\bar{R}}$, where[8] $M_{\bar{R}} := \max\limits_{\|x - x^*\| \leq 2\bar{R}} \|\nabla f(x)\|$. Then Algorithm 1 guarantees that*

$$f(y^{k+1}) - f(x^*) \leq \Gamma_{k+1}\bar{R}^2 \leq \frac{18\ell(0)\bar{R}^2}{\left(k + 1 - \bar{k}\right)^2}$$

*for all $k \geq \bar{k} := \max\left\{1 + \frac{1}{2}\log_{3/2}\left(\frac{\Gamma_0}{8\ell(0)}\right), 0\right\}$ with any $\bar{R} \geq \|x^0 - x^*\|$.*

*Proof.* In our proof, we define the Lyapunov function $V_k := f(y^k) - f(x^*) + \frac{\Gamma_k}{2}\|u^k - x^*\|^2$.

*(Base case:)* Clearly, $\|u^0 - x^*\| = \|y^0 - x^*\| \leq \|x^0 - x^*\| \leq 2\bar{R}$ due the the monotonicity of GD (Tyurin, 2025)[Lemma I.2] and $\bar{R} \geq R$. Thus,

$$\left\|\nabla f(y^0)\right\| \leq \max\limits_{\|x - x^*\| \leq 2\bar{R}} \|\nabla f(x)\| \leq M_{\bar{R}}.$$

Using Lemma B.4, either $\left\|\nabla f(y^0)\right\| \leq \Delta_{\text{left}}(\delta)$ or $\left\|\nabla f(y^0)\right\| \geq \Delta_{\text{right}}(\delta)$. However, the latter is not possible because $\Delta_{\text{right}}(\delta) > M_{\bar{R}}$ and $\left\|\nabla f(y^0)\right\| \leq M_{\bar{R}}$. Thus, $\ell\left(4\left\|\nabla f(y^0)\right\|\right) \leq \ell(4\Delta_{\text{left}}(\delta)) \leq 2\ell(0)$, where the last inequality due to the conditions of the theorem.

Trivially, $V_0 \leq V_0$ and

$$V_0 = f(y^0) - f(x^*) + \frac{\Gamma_0}{2}\|y^0 - x^*\|^2 \leq \frac{\delta}{2} + \frac{\Gamma_0}{2}\|y^0 - x^*\|^2$$

$$\leq \frac{\delta}{2} + \frac{\Gamma_0}{2}\|x^0 - x^*\|^2 \leq \delta \tag{43}$$

since $\Gamma_0 = \frac{\delta}{\bar{R}^2}$ and $\bar{R} \geq \|x^0 - x^*\|$. Using mathematical induction, we assume that $\ell\left(4\left\|\nabla f(y^k)\right\|\right) \leq 2\ell(0)$,

$$V_k \leq \left(\prod_{i=0}^{k-1}\frac{1}{1 + \alpha_i}\right)V_0, \tag{44}$$

$\left\|u^k - x^*\right\| \leq 2\bar{R}$, and $\left\|y^k - x^*\right\| \leq 2\bar{R}$ for some $k \geq 0$ (the base case has been proved in the previous steps).

Consider Lemma D.1 and the steps (26). Then,

$$(1 + \alpha_{k,\gamma})(f(y_\gamma^{k+1}) - f(x^*)) + \frac{(1 + \alpha_{k,\gamma})\Gamma_{k+1,\gamma}}{2}\left\|u_\gamma^{k+1} - x^*\right\|^2 - \left((f(y^k) - f(x^*)) + \frac{\Gamma_k}{2}\left\|u^k - x^*\right\|^2\right)$$

$$\leq \frac{1}{2}\left(\gamma - \frac{1}{\ell(2\left\|\nabla f(y^k)\right\| + \left\|\nabla f(y_\gamma^{k+1})\right\|)}\right)\left\|\nabla f(y_\gamma^{k+1}) - \nabla f(y^k)\right\|^2,$$

where $0 \leq \gamma \leq \frac{1}{\ell(2\|\nabla f(y^k)\|)}$ is a free parameter. Let us take the smallest $\gamma$ such that

$$g(\gamma) := \gamma - \frac{1}{\ell(2\left\|\nabla f(y^k)\right\| + \left\|\nabla f(y_\gamma^{k+1})\right\|)} = 0$$

---

[7]if the set $\{x \in [\Delta_{\max}, \infty) : \psi(x) = \delta\}$ is empty, then $\Delta_{\text{right}}(\delta) = \infty$

[8]or is it sufficient to find any $M_{\bar{R}}$ such that $M_{\bar{R}} \geq \max\limits_{\|x - x^*\| \leq 2\bar{R}} \|\nabla f(x)\|$.

and denote is as $\gamma^*$ (exists similarly to the proof of Theorem 3.2 and $\gamma^* \leq \frac{1}{\ell(2\|\nabla f(y^k)\|)}$). For all $\gamma \leq \gamma^*$, $g(\gamma) \leq 0$ and

$$
\begin{aligned}
(1 + \alpha_{k,\gamma})(f(y_\gamma^{k+1}) - f(x^*)) + \frac{(1 + \alpha_{k,\gamma})\Gamma_{k+1,\gamma}}{2} \left\| u_\gamma^{k+1} - x^* \right\|^2 \\
\leq (f(y^k) - f(x^*)) + \frac{\Gamma_k}{2} \left\| u^k - x^* \right\|^2 =: V_k,
\end{aligned}
\tag{45}
$$

which ensures that

$$
f(y_\gamma^{k+1}) - f(x^*) \leq V_k \overset{(44)}{\leq} V_0 \overset{(43)}{\leq} \delta.
\tag{46}
$$

Moreover, due to (45) and (26), we have

$$
\begin{aligned}
\frac{\Gamma_k}{2} \left\| u_\gamma^{k+1} - x^* \right\|^2 = \frac{(1 + \alpha_{k,\gamma})\Gamma_{k+1,\gamma}}{2} \left\| u_\gamma^{k+1} - x^* \right\|^2 \leq V_k \\
\overset{(44)}{\leq} \left( \prod_{i=0}^{k-1} \frac{1}{1 + \alpha_i} \right) V_0 = \frac{\Gamma_k}{\Gamma_0} \left( (f(y^0) - f(x^*)) + \frac{\Gamma_0}{2} \left\| u^0 - x^* \right\|^2 \right) \\
\overset{\text{Alg. 1}}{\leq} \Gamma_k \left( \frac{\delta}{2\Gamma_0} + \frac{1}{2} \left\| u^0 - x^* \right\|^2 \right) \leq \Gamma_k \bar{R}^2,
\end{aligned}
$$

where the last inequality due to $\Gamma_0 = \frac{\delta}{\bar{R}^2}$ and $\left\| u^0 - x^* \right\|^2 \leq \bar{R}^2$. Thus,

$$
\left\| u_\gamma^{k+1} - x^* \right\|^2 \leq 2\bar{R}
\tag{47}
$$

for all $\gamma \leq \gamma^*$. Now, consider $y_\gamma^{k+1}$ from (26):

$$
\begin{aligned}
& \left\| y_\gamma^{k+1} - x^* \right\| \\
&= \left\| \frac{1}{1 + \alpha_{k,\gamma}} y^k + \frac{\alpha_{k,\gamma}}{1 + \alpha_{k,\gamma}} u^k - \frac{\gamma}{1 + \alpha_{k,\gamma}} \nabla f(y^k) - x^* \right\| \\
&= \left\| \frac{1}{1 + \alpha_{k,\gamma}} \left( (y^k - \gamma \nabla f(y^k)) - x^* \right) + \frac{\alpha_{k,\gamma}}{1 + \alpha_{k,\gamma}} (u^k - x^*) \right\| \\
&\leq \frac{1}{1 + \alpha_{k,\gamma}} \left\| (y^k - \gamma \nabla f(y^k)) - x^* \right\| + \frac{\alpha_{k,\gamma}}{1 + \alpha_{k,\gamma}} \left\| u^k - x^* \right\|,
\end{aligned}
\tag{48}
$$

where we use Triangle's inequality. Notice that

$$
\gamma \leq \frac{1}{\ell(2\|\nabla f(y^k)\|)}
\tag{49}
$$

for all $\gamma \leq \gamma^*$ because $\gamma^* \leq \frac{1}{\ell(2\|\nabla f(y^k)\|)}$. Thus,

$$
\begin{aligned}
\left\| (y^k - \gamma \nabla f(y^k)) - x^* \right\|^2 &= \left\| y^k - x^* \right\|^2 - 2\gamma \langle y^k - x^*, \nabla f(y^k) \rangle + \gamma^2 \left\| \nabla f(y^k) \right\|^2 \\
&\overset{\text{L. B.1}}{\leq} \left\| y^k - x^* \right\|^2 + 2\gamma \left( f(x^*) - f(y^k) - \left\| \nabla f(y^k) \right\|^2 \int_0^1 \frac{1 - v}{\ell(\|\nabla f(y^k)\| v)} dv \right) + \gamma^2 \left\| \nabla f(y^k) \right\|^2 \\
&\leq \left\| y^k - x^* \right\|^2 + \gamma \left\| \nabla f(y^k) \right\|^2 \left( \gamma - 2 \int_0^1 \frac{1 - v}{\ell(\|\nabla f(y^k)\| v)} dv \right).
\end{aligned}
$$

In the last inequality, we use $f(x^*) - f(y^k) \leq 0$. Next,

$$
\begin{aligned}
\left\| (y^k - \gamma \nabla f(y^k)) - x^* \right\|^2 &\overset{(49)}{\leq} \left\| y^k - x^* \right\|^2 + \gamma \left\| \nabla f(y^k) \right\|^2 \left( \frac{1}{\ell(2\|\nabla f(y^k)\|)} - 2 \int_0^1 \frac{1 - v}{\ell(\|\nabla f(y^k)\| v)} dv \right) \\
&\leq \left\| y^k - x^* \right\|^2 + \gamma \left\| \nabla f(y^k) \right\|^2 \left( \frac{1}{\ell(2\|\nabla f(y^k)\|)} - \frac{1}{\ell(\|\nabla f(y^k)\|)} \right) \\
&\leq \left\| y^k - x^* \right\|^2
\end{aligned}
$$

because $\ell$ is non-decreasing. Thus, by the induction assumption, $\left\| \left( y^k - \gamma \nabla f(y^k) \right) - x^* \right\| \leq \left\| y^k - x^* \right\| \leq 2\bar{R}$, $\left\| u^k - x^* \right\| \leq 2\bar{R}$, and

$$\left\| y_\gamma^{k+1} - x^* \right\| \leq 2\bar{R} \tag{50}$$

for all $\gamma \leq \gamma^*$, due to (48).

Thus,

$$\left\| \nabla f(y_\gamma^{k+1}) \right\| \leq \max_{\|x - x^*\| \leq 2\bar{R}} \|\nabla f(x)\| \leq M_{\bar{R}}.$$

Using (46) and Lemma B.4, either $\left\| \nabla f(y_\gamma^{k+1}) \right\| \leq \Delta_{\text{left}}(\delta)$ or $\left\| \nabla f(y_\gamma^{k+1}) \right\| \geq \Delta_{\text{right}}(\delta)$. However, the latter is not possible because $\Delta_{\text{right}}(\delta) > M_{\bar{R}}$ and $\left\| \nabla f(y_\gamma^{k+1}) \right\| \leq M_{\bar{R}}$. Thus,

$$\ell\left( 4 \left\| \nabla f(y_\gamma^{k+1}) \right\| \right) \leq \ell(4\Delta_{\text{left}}(\delta)) \leq 2\ell(0). \tag{51}$$

Therefore, by the definition of $\gamma^*$ and using $\ell\left( 4 \left\| \nabla f(y^k) \right\| \right) \leq 2\ell(0)$,

$$\gamma^* = \frac{1}{\ell(2 \|\nabla f(y^k)\| + \|\nabla f(y_{\gamma^*}^{k+1})\|)} \geq \frac{1}{\max\{\ell(4 \|\nabla f(y^k)\|), \ell(4 \|\nabla f(y_{\gamma^*}^{k+1})\|)\}} \geq \frac{1}{2\ell(0)},$$

meaning that we can take $\gamma = \frac{1}{2\ell(0)}$ and (45) holds:

$$(1 + \alpha_{k,\gamma})(f(y_\gamma^{k+1}) - f(x^*)) + \frac{(1 + \alpha_{k,\gamma})\Gamma_{k+1,\gamma}}{2} \left\| u_\gamma^{k+1} - x^* \right\|^2 \leq V_k.$$

Notice that $\alpha_{k,\gamma} = \alpha_k$, $y_\gamma^{k+1} = y^{k+1}$, $\Gamma_{k+1,\gamma} = \Gamma_{k+1}$, and $u_\gamma^{k+1} = u^{k+1}$ with $\gamma = \frac{1}{2\ell(0)}$. Therefore,

$$(1 + \alpha_{k,\gamma})(f(y_\gamma^{k+1}) - f(x^*)) + \frac{(1+\alpha_{k,\gamma})\Gamma_{k+1,\gamma}}{2} \left\| u_\gamma^{k+1} - x^* \right\|^2 = (1 + \alpha_k)V_{k+1},$$

$$\ell\left( 4 \left\| \nabla f(y^{k+1}) \right\| \right) \overset{(51)}{\leq} 2\ell(0),$$

$$V_{k+1} \leq \frac{1}{1 + \alpha_k} V_k \leq \left( \prod_{i=0}^{k} \frac{1}{1 + \alpha_i} \right) V_0,$$

$$\left\| u^{k+1} - x^* \right\|^2 \overset{(47)}{\leq} 2\bar{R},$$

and

$$\left\| y^{k+1} - x^* \right\| \overset{(50)}{\leq} 2\bar{R}.$$

We have proved the next step of the induction. Finally, for all $k \geq 0$,

$$f(y^{k+1}) - f(x^*) \leq V_{k+1} \leq \left( \prod_{i=0}^{k} \frac{1}{1 + \alpha_i} \right) \left( f(y^0) - f(x^*) + \frac{\Gamma_0}{2} \left\| y^0 - x^* \right\|^2 \right)$$

$$\leq \Gamma_0 \left( \prod_{i=0}^{k} \frac{1}{1 + \alpha_i} \right) \left( \frac{\delta}{2\Gamma_0} + \frac{1}{2} \left\| y^0 - x^* \right\|^2 \right) \leq \Gamma_{k+1} \bar{R}^2$$

because GD by (Tyurin, 2025)[Lemma I.2] returns $\bar{x} = y^0$ such that $\left\| y^0 - x^* \right\| \leq \left\| x^0 - x^* \right\| \leq \bar{R}$. Moreover, we use $\Gamma_0 = \frac{\delta}{\bar{R}^2}$ and $\Gamma_{k+1} = \Gamma_0 \left( \prod_{i=0}^{k} \frac{1}{1+\alpha_i} \right)$. It is left to use Theorem 3.1. $\qquad \square$

**Theorem 5.2.** *Consider the assumptions and results of Theorem 5.1. The oracle complexity (i.e., the number of gradient calls) required to find an $\varepsilon$–solution is*

$$\frac{5\sqrt{\ell(0)}\bar{R}}{\sqrt{\varepsilon}} + k(\delta)$$

*for all $\delta \in Q$, where $k(\delta) := \max\left\{ 1 + \frac{1}{2} \log_{3/2} \left( \frac{\delta}{8\ell(0)\bar{R}^2} \right), 0 \right\} + k_{\text{GD}}(\delta)$, $k_{\text{GD}}(\delta)$ is the oracle complexity of GD for finding a point $\bar{x}$ such that $f(\bar{x}) - f(x^*) \leq \delta/2$.*

*Proof.* The proof of this theorem repeats the proof of Theorem 3.3, with the only change being that the conditions on $\delta$ are different. $\qquad \square$

### E.3.1 EXAMPLE: $(\rho, L_0, L_1)$–SMOOTHNESS

To explain how Theorem 5.2 and Corollary 5.3 work, let us consider $(\rho, L_0, L_1)$–smoothness with $\ell(x) = L_0 + L_1 x^\rho$ and $\rho > 0$. In this case, $\psi(x) \simeq \frac{x^2}{L_0 + L_1 x^\rho}$, which is strictly increasing until $\Delta_{\max} = \infty$ if $\rho \leq 2$, and until $\Delta_{\max} = (2L_0/((\rho - 2)L_1))^{1/\rho}$ if $\rho > 2$. If $\rho \leq 2$, then $Q := \left\{\delta \geq 0 : \ell(4\psi^{-1}(\delta)) \leq 2\ell(0)\right\} = \left\{\delta \geq 0 : \ell(8\sqrt{\delta\ell(0)}) \leq 2\ell(0)\right\} = \left\{\delta \geq 0 : \delta \leq L_0^{2/\rho - 1}/(64 L_1^{2/\rho})\right\}$ and, using the result from Table 2 by Tyurin (2025) with $\rho < 2$ and Theorem 5.2,

$$\frac{5\sqrt{\ell(0)}\bar{R}}{\sqrt{\varepsilon}} + \min_{\delta \in Q} k(\delta)$$

$$= \mathcal{O}\left(\frac{\sqrt{L_0}\bar{R}}{\sqrt{\varepsilon}} + \min_{\delta \in Q}\left[\max\left\{\log\left(\frac{\delta}{L_0\bar{R}^2}\right), 0\right\} + \frac{L_0\bar{R}^2}{\delta} + \frac{L_1\Delta^{\rho/2}\bar{R}^{2-\rho}}{\delta^{1-\rho/2}}\right]\right)$$

$$= \mathcal{O}\left(\frac{\sqrt{L_0}R}{\sqrt{\varepsilon}} + \frac{L_1\Delta^{\rho/2}}{L_0^{1-\rho/2}} + L_1^{2/\rho}L_0^{2-2/\rho}R^2 + \frac{L_1^{2/\rho}\Delta^{\rho/2}R^{2-\rho}}{L_0^{2/\rho + \rho/2 - 2}}\right).$$

where $\Delta := f(x^0) - f(x^*)$, and we take $\bar{R} = R$ and $\delta = \min\{L_0^{2/\rho - 1}/L_1^{2/\rho}, L_0\bar{R}^2\}/64$ to get the last complexity (which might not be the optimal choice, but a sufficient choice to show that the first term dominates if $\varepsilon$ is small). Similarly, for the case $\rho = 2$, the oracle complexity at least

$$\mathcal{O}\left(\frac{\sqrt{L_0}R}{\sqrt{\varepsilon}} + \frac{L_0 R^2}{\bar{\delta}} + \frac{L_1 M_0^\rho R^2}{\bar{\delta}}\right)$$

with $\bar{\delta} = \min\{L_0^{2/\rho - 1}/L_1^{2/\rho}, L_0 R^2\}/64$ and $\bar{R} = R$, where we take the GD rate from (Li et al., 2024a; Tyurin, 2025).

We now consider the case $\rho > 2$. Let us define $\Delta_1 := \frac{1}{2}(L_0/L_1)^{1/\rho}$. Notice that $\Delta_{\max} \geq \Delta_1$. For all $\delta \in [0, \psi(\Delta_1))$, we can find $\Delta_{\text{left}}(\delta) = \psi^{-1}(\delta) \simeq \sqrt{L_0\delta}$. For all $x \geq \Delta_{\max}$, $\psi(x)$ is decreasing, and $\psi(x) \simeq \frac{x^2}{L_1 x^\rho}$ Thus, $\Delta_{\text{right}}(\delta) \simeq (L_1\delta)^{1/(2-\rho)}$ and we should minimize $k(\delta)$ over the set $\{\delta \in [0, L_0^{2/\rho - 1}/L_1^{2/\rho}] : \delta \leq L_0/L_1^2, \delta \leq (1/(2M_{\bar{R}}))^{\rho-2}/L_1\} \subseteq Q$ (up to constant factors). It is sufficient to take

$$\bar{\delta} := \min\{L_0^{2/\rho - 1}/L_1^{2/\rho}, L_0/L_1^2, (1/(2M_{\bar{R}}))^{\rho - 2}/L_1, L_0\bar{R}^2\} \tag{52}$$

to get the complexity

$$\mathcal{O}\left(\frac{\sqrt{L_0}R}{\sqrt{\varepsilon}} + \min_{\delta \in Q} k(\delta)\right) = \mathcal{O}\left(\frac{\sqrt{L_0}R}{\sqrt{\varepsilon}} + \frac{L_0 R^2}{\bar{\delta}} + \frac{L_1 M_0^\rho R^2}{\bar{\delta}}\right),$$

where $M_0 := \|\nabla f(x^0)\|$, $k_{\text{GD}}(\delta)$ is derived using (Li et al., 2024a; Tyurin, 2025), and we take $\bar{R} = R$. Thus, we can guarantee the $\sqrt{L_0}R/\sqrt{\varepsilon}$ rate for any $\rho \geq 0$ and a sufficiently small $\varepsilon$.

