# OpenReview forum: "Near-Optimal Convergence of Accelerated Gradient Methods under Generalized and $(L_0, L_1)$-Smoothness"
_ICLR.cc/2026/Conference — Submitted to ICLR 2026_

### Official Review · Reviewer_MkRn · 2025-10-30

**Soundness:** 3
**Presentation:** 2
**Contribution:** 2
**Rating:** 4
**Confidence:** 4

**Summary:**

This paper presents a new accelerated gradient descent method for convex optimization under a generalized $\ell$-smoothness assumption, establishing an AGD-type complexity that improves upon existing convergence guarantees.

**Strengths:**

1. **Originality:** Building on a previously proposed assumption and existing methods, the paper establishes an improved complexity bound.

2. **Quality:** While we have concerns about the writing clarity (discussed in *Weaknesses*), the overall structure is sound and the presentation is generally clear.

3. **Clarity:** The paper clearly explains its motivation and main ideas.

4. **Significance:** The new result may advance $\ell$-smooth optimization theory. However, given the limited empirical advantage over GD-based methods, it remains unclear whether the method achieves practical acceleration.

**Weaknesses:**

1. The proof technique is not novel. The proposed method is a discretization of a rotated heavy-ball flow introduced in [1]. With a simple change of variables, the Lyapunov function coincides with those in [2], [3], or [4]. Modern continuous-flow–based proofs, such as [5] and [6], could simplify the argument substantially and shorten Appendix B.

2. Lines 272–274 state: “While for $L$–smooth functions the proof technique from (Wei & Chen, 2025) does not offer any advantages over, for example, (Nesterov, 1983) because the result in (Nesterov, 1983) is optimal.” However, (Nesterov, 1983) is not optimal in the information-theoretic sense. There is extensive work on optimal gradient methods (OGM); see, for example, [7].

3. The numerical experiments in Appendix A do not demonstrate the efficiency of the proposed method. The AGD curve shows no improvement over GD, contradicting the theoretical claims. In addition, AGD exhibits severe oscillations, suggesting that it may not achieve genuine acceleration—or even reliable convergence.

**References**

[1] Wei, J., & Chen, L. (2024). Accelerated Over-Relaxation Heavy-Ball Method: Achieving Global Accelerated Convergence with Broad Generalization. *ICLR*, 2025.

[2] Alvarez, F., & Attouch, H. (2001). An inertial proximal method for maximal monotone operators via discretization of a nonlinear oscillator with damping. Set-Valued Analysis, 9(1–2), 3–11. https://doi.org/10.1023/A:1011203001547

[3] Su, W., Boyd, S., & Candès, E. J. (2016). A differential equation for modeling Nesterov’s accelerated gradient method: Theory and insights. *Journal of Machine Learning Research*, 17(153), 1–43. http://jmlr.org/papers/v17/15-084.html

[4] Wibisono, Wilson, & Jordan (2016). A variational perspective on accelerated methods in optimization. *Proceedings of the National Academy of Sciences*, 113(47), E7351–E7358. https://doi.org/10.1073/pnas.1614734113

[5] Chen, L., & Luo, H. (2019). First-order optimization methods based on Hessian-driven Nesterov accelerated gradient flow. arXiv preprint arXiv:1912.09276.

[6] Luo, H., & Chen, L. (2022). From differential equation solvers to accelerated first-order methods for convex optimization. *Mathematical Programming*, 195(1), 735-781. https://doi.org/10.1007/s10107-021-01713-3

[7] Kim, D., Fessler, J.A. (2015). Optimized first-order methods for smooth convex minimization. *Mathematical Programming*. https://10.1007/s10107-015-0949-3

**Questions:**

1. Beyond the generalized smoothness assumption, is there an analogous assumption that extends strong convexity? Under such a condition, can the method achieve linear convergence?

2. What convergence guarantees can be established when applying gradient methods to $\ell$-smooth functions without assuming convexity?

---

> ### Author Response · Authors · 2025-11-13
>
> Thank you. Notice that we've added new experiments into the PDF (Section A). Let us respond to the weaknesses and questions.
>
> > The proof technique is not novel. The proposed method is a discretization of a rotated heavy-ball flow introduced in [1]. With a simple change of variables, the Lyapunov function coincides with those in [2], [3], or [4]. Modern continuous-flow–based proofs, such as [5] and [6], could simplify the argument substantially and shorten Appendix B.
>
> We believe that our Sections 3.2 and 3.3 are overlooked. There we explain that the proof techniques require new tricks and tools, which are novel and non-trivial.
>
> > Lines 272–274 state: “While for L–smooth functions the proof technique from (Wei & Chen, 2025) does not offer any advantages over, for example, (Nesterov, 1983) because the result in (Nesterov, 1983) is optimal.” However, (Nesterov, 1983) is not optimal in the information-theoretic sense. There is extensive work on optimal gradient methods (OGM); see, for example, [7].
>
> If you are asking about improving the universal constant factor $C$ near the term $C \times \sqrt{\frac{L R^2}{\varepsilon}}$ in (Nesterov, 1983), then indeed $C$ can be improved, but (Nesterov, 1983) is optimal up to $C$ and for large dimension $d$ (see Nesterov, 2018). In our paper, we compare the complexities up to universal constant factors (e.g., $1, 3, 8$). Our work improves a non-constant factor (e.g., $L_0$ and $L_1$), which is more important. Improving the universal constant factor is the next natural research step.
>
> > The numerical experiments in Appendix A do not demonstrate the efficiency of the proposed method. The AGD curve shows no improvement over GD, contradicting the theoretical claims.
>
> We respectfully disagree. AGD’s curve lies below GD after a few iterations in Section A, which aligns with our theory.
>
> > In addition, AGD exhibits severe oscillations, suggesting that it may not achieve genuine acceleration—or even reliable convergence.
>
> The oscillations are a natural behavior of accelerated methods, which can be observed not only in our experiments but also in [1,2].
>
> > Beyond the generalized smoothness assumption, is there an analogous assumption that extends strong convexity?
>
> We are not aware of such an assumption.
>
> > Under such a condition, can the method achieve linear convergence?
>
> Under the standard strong convexity, one might apply the "restart technique."
>
> > What convergence guarantees can be established when applying gradient methods to $\ell$-smooth functions without assuming convexity?
>
> Our work develops accelerated methods and requires convex functions for that. For non-convex analysis, we refer to previous works [1-3].
>
> **We believe that we have responded to all the questions. If you have more questions, please let us know.**
>
> ---
>
> [1] Eduard Gorbunov, Nazarii Tupitsa, Sayantan Choudhury, Alen Aliev, Peter Richt´arik, Samuel Horv´ath, and Martin Tak´aˇc. Methods for convex (L0,L1)-smooth optimization: Clipping, acceleration, and adaptivity. In International Conference on Learning Representations, 2025.
>
> [2] Daniil Vankov, Anton Rodomanov, Angelia Nedich, Lalitha Sankar, and Sebastian U Stich. Optimizing (L0,L1)-smooth functions by gradient methods. arXiv preprint arXiv:2410.10800, 2024.
>
> [3] Alexander Tyurin Toward a unified theory of gradient descent under generalized smoothness. ICLR 2025

---

> > ### Comment · Reviewer_MkRn · 2025-11-22
> >
> > We admit that the way to choose the reciprocal of the local smoothness constant (i.e. $\gamma$) is new and special in the generalized smoothness setting.
> >
> > However, some of the authors' claims in the section remain, in my view, not entirely appropriate. The choice of Lyapunov function coincides with prior literature, for example, Equation (74) of [Luo and Chen 2022]; Theorem 3 of [Taylor and Drori 2022]; Equation (21) of [Chen and Luo 2019]. Thus, the authors' claims in Line 279-282
> >
> > > This stands in contrast to all known proofs we are aware of, which typically use a different point, one
> > where the gradient is not evaluated.
> >
> > should be reconsidered.
> >
> > About the novelty of the proof, please check Lemma 8.1 in  **A Unified Convergence Analysis of First Order Convex Optimization Methods via Strong Lyapunov Functions**. Long Chen, Hao Luo. arXiv, and Theorem 3.1 and 4.1 in **First order optimization methods based on Hessian-driven Nesterov accelerated gradient flow**. Long Chen, Hao Luo.  arXiv.

---

> > ### Comment · Reviewer_MkRn · 2025-11-22
> >
> > > We respectfully disagree. AGD’s curve lies below GD after a few iterations in Section A, which aligns with our theory.
> > > The oscillations are a natural behavior of accelerated methods, which can be observed not only in our experiments but also in [1,2].
> > Oscillations can indeed occur in accelerated methods, but I still have several concerns:
> >
> > The oscillations in your plots are significantly larger than those typically observed in [1,2], which casts doubt on the robustness of the algorithm under the chosen parameterization.
> >
> > The error curve is essentially linear, which does not match the accelerated rate predicted by your theorems. This may be due to the fact that the objective function is strongly convex, which is also why I raised the question about linear convergence.
> >
> > Even ignoring the oscillations, the convergence rate of AGD appears nearly identical to that of GD in your experiments. This makes it difficult to see the empirical effect of acceleration.

---

> ### Author Response · Authors · 2025-11-22
>
> > However, some of the authors' claims in the section remain, in my view, not entirely appropriate. The choice of Lyapunov function coincides with prior literature, for example, Equation (74) of [Luo and Chen 2022]; Theorem 3 of [Taylor and Drori 2022]; Equation (21) of [Chen and Luo 2019]. Thus, the authors' claims in Line 279-282 "This stands in contrast to all known proofs we are aware of, which typically use a different point, one where the gradient is not evaluated. "should be reconsidered.
>
> This part was misunderstood. Of course, the Lyapunov function is the same as in (Wei & Chen, 2025), and we do not hide it; the entire Section 3.2 explains this. Nevertheless, we have removed this sentence and also revised the abstract and the paper (see the updated pdf). We don't claim that the algorithms are entirely new.
>
> > About the novelty of the proof, please check Lemma 8.1 in A Unified Convergence Analysis of First Order Convex Optimization Methods via Strong Lyapunov Functions. Long Chen, Hao Luo. arXiv, and Theorem 3.1 and 4.1 in First order optimization methods based on Hessian-driven Nesterov accelerated gradient flow. Long Chen, Hao Luo. arXiv.
>
> **The referenced papers analyze the standard $L$–smoothness setting, so their results are not relevant to us. Adapting the previous proofs from the standard $L$–smoothness setting to the generalized one is a non-trivial task. At least three different teams were unable to adapt the previous techniques and obtain the optimal time complexity (see Table 1). Our work is the first to do so.**

---

> ### Author Response · Authors · 2025-11-22
>
> > The oscillations in your plots are significantly larger than those typically observed in [1,2], which casts doubt on the robustness of the algorithm under the chosen parameterization. The error curve is essentially linear, which does not match the accelerated rate predicted by your theorems. This may be due to the fact that the objective function is strongly convex, which is also why I raised the question about linear convergence.
>
> In the updated pdf, we also implemented the method by [1], and the frequency of oscillations of both methods is the same.
>
> > Even ignoring the oscillations, the convergence rate of AGD appears nearly identical to that of GD in your experiments. This makes it difficult to see the empirical effect of acceleration.
>
> In Figure 2, the convergence of our method and of the method in [1] is much faster. The curves of the accelerated methods are significantly lower. In Figure 1, our method converges to $10^{-9}$ after 20K iterations, while GD only converges to $10^{-6}.$
>
> ---
>
> Nevertheless, this is a theoretical work that provides important new results for the optimization community in the generalized smoothness setting. Even without experiments (which we actually have), the contribution remains strong.

---

### Official Review · Reviewer_XLXR · 2025-10-30

**Soundness:** 3
**Presentation:** 2
**Contribution:** 2
**Rating:** 4
**Confidence:** 3

**Summary:**

The paper propose first-order methods for convex optimization problems with $\ell$-smoothness condition: $\|\nabla ^2 f(x)\| \leq \ell(\|\nabla f(x)\|)$, where $\ell$ is a non-decreasing, positive, locally Lipschitz function. When $\psi(x) = \frac{x^2}{2\ell(4x)}$ is strictly increasing, the proposed algorithms achieve the oracle complexity of $O(\sqrt{\ell(0)}R/ \sqrt{\varepsilon})$ for $R = \|x_0 - x^*\|$ and small $\varepsilon$. In particualr, with $(L_0, L_1)$-smoothness, the oracle complexity is $O(\sqrt{L_0}R/{\sqrt{\varepsilon}})$.

**Strengths:**

-  The proposed algorithms improve the the oracle complexity of accerated gradient methods on a class of convex optimization problems over $\ell$-smoothness condition.

- The proofs of the main theorems are sound and well-discussed.

**Weaknesses:**

- The assumption that $\psi(x)$ is strictly increasing restricts the results mainly to $(L_0, L_1)$-smoothness.


- The algorithms requires sophiticated choices on parameters which are usually unknown or difficult to estimate; while the optimal convegence region relies on these parameters.

**Questions:**

- What is the motivation of considering the function $\psi(x)$?

- Beyond $(L_0, L_1)$-smoothness, what are the functions $\ell$ such that the assumption $\psi(x)$ is strictly increasing holds?


- For Algorithm 1, what is the convegence guarantee after GD and before $\bar k$ itertations?


- There is a non-accelerated phase in Algorithm 1 and results in addtional constant factors in the oracle complexity. The acclerated region relies on $\delta$ and $\bar R$.  Should this be viewed as local accelerated convergence (requiring initial guess close to the solution)?

- The paper claims the convergence rate is a significant improvement over previous works. It would be nice to have some numerical experiments compared with other AGD methods to validate the performance of the proposed algorithms.

- Is the set $Q$ defined in line 441 non-empty?

---

> ### Author Response · Authors · 2025-11-13
>
> Thank you. Notice that we've added new experiments into the PDF (Section A).
>
> > The assumption that $\psi$ is strictly increasing restricts the results mainly to $(L_0, L_1)$-smoothness.
>
> We **don’t** use this assumption across the whole paper. For instance, in Section 5 and E.3.1, the theorem works even if $\psi$ is not strictly increasing. Note that Alg. 1 supports any $\psi$.
>
> > The algorithms requires sophiticated choices on parameters which are usually unknown or difficult to estimate; while the optimal convegence region relies on these parameters.
>
> We've added one more column to Table 1 of the PDF. Different methods provide different trade-offs. For instance, AGD by (Gorbunov et al.) requires fewer parameters, but it has an exponential dependence on them. On the other hand, our method requires four parameters (and is semi-adaptive to two of them), but it is optimal for small $\varepsilon$.
>
> > What is the motivation of considering the function $\psi$?
>
> The function $\psi$ naturally appears everywhere in the paper. We define $\psi$ to improve readability, since $\psi$ appears in many places.
>
> > Beyond $(L_0, L_1)$-smoothness, what are the functions $\ell$ such that the assumption $\psi(x)$ is strictly increasing holds?
>
> We don’t need the assumption that $\psi$ is strictly increasing. In Section 5, the theorem works even if $\psi$ is not strictly increasing. In Section E.3.1, we consider an example when $\psi$ is neither increasing nor decreasing.
>
> > For Algorithm 1, what is the convegence guarantee after GD and before itertations?
>
> The convergence rate can be found in [1–4]. In Line 238, we specify the exact formula.
>
> > There is a non-accelerated phase in Algorithm 1 and results in addtional constant factors in the oracle complexity. The acclerated region relies on. Should this be viewed as local accelerated convergence (requiring initial guess close to the solution)?
>
> In some sense, yes. Algorithm 1 is only locally accelerated, which is sufficient for the optimal oracle complexity for small $\varepsilon$. However, Algorithm 2 is globally accelerated (unlike Algorithm 1).
>
> > The paper claims the convergence rate is a significant improvement over previous works. It would be nice to have some numerical experiments compared with other AGD methods to validate the performance of the proposed algorithms.
>
> We have added an experiment to Section A where we compare other AGD methods with ours, and we also include a sensitivity-to-parameters study. All experiments are consistent with the theoretical results. The difference can be huge. In the case of $(L_0, L_1)$–smoothness, the ratio is $\frac{\ell(||\nabla f(x_0)||)}{\ell(0)} = \frac{L_0 + L_1 ||\nabla f(x_0)||}{L_0}$ compared to [1]. The norm can be as large as $\Theta(e^{L_1 R})$ for the function $e^{L_1 R}$. Thus, the difference is exponential!
>
> > Is the set $Q$ defined in line 441 non-empty?
>
> Yes, it is non-empty due to Lemma B.4. See an example in Section E.3.1.
>
> **We believe that we addressed all the questions. If you have more questions, please let us know.**
>
> ---
>
> [1] Haochuan Li, Jian Qian, Yi Tian, Alexander Rakhlin, and Ali Jadbabaie. Convex and non-convex optimization under generalized smoothness. Advances in Neural Information Processing Systems, 36, 2024a.
>
> [2] Eduard Gorbunov, Nazarii Tupitsa, Sayantan Choudhury, Alen Aliev, Peter Richt´arik, Samuel Horv´ath, and Martin Tak´aˇc. Methods for convex (L0,L1)-smooth optimization: Clipping, acceleration, and adaptivity. In International Conference on Learning Representations, 2025.
>
> [3] Daniil Vankov, Anton Rodomanov, Angelia Nedich, Lalitha Sankar, and Sebastian U Stich. Optimizing (L0,L1)-smooth functions by gradient methods. arXiv preprint arXiv:2410.10800, 2024.
>
> [4] Alexander Tyurin Toward a unified theory of gradient descent under generalized smoothness. ICLR 2025

---

> ### Comment · Reviewer_XLXR · 2025-11-19
>
> Thank you for highlighting example E.3.1. Did you test on any example that $\psi$ is not strictly increasing?
>
> > The convergence rate can be found in [1–4]. In Line 238, we specify the exact formula.
>
> The question was regrading to the initial phase of Algorithm 1 **after GD** as there seems no convergence guarantee for $k \leq \bar k$ and $\bar k$ is related to the choice of parameters. Can you justify that (accelerated) convergence is not restricted to a small region, for example is $\bar k$ usually a small number close to 0?

---

> ### Author Response · Authors · 2025-11-19
>
> Thank you for the reponse.
>
> > Did you test on any example that $\psi$ is not strictly increasing?
>
> Yes, see our example in Section E.3.1.
>
> > The question was regrading to the initial phase of Algorithm 1 after GD as there seems no convergence guarantee for $k \leq \bar k$ and $\bar k$ is related to the choice of parameters. Can you justify that (accelerated) convergence is not restricted to a small region, for example is $\bar k$ usually a small number close to 0?
>
> In this theory, there are no guarantees for $k < \bar k.$ So, we should wait at least $\bar k$ iterations to guarantee the convergence. However, $\bar k$ is tiny since $\bar k = \Theta(\log \left(\frac{\delta}{8 \ell(0) \bar{R}^2}\right))$ (it has the logarithmic dependence on the parameters). The waiting time is negligible.
>
> For instance, in the case of $(L_0, L_1)$-smoothness, $\bar{k} = \Theta(\min[\log \left(\frac{1}{L_1^2 R^2}\right), 1])$ for our choice of $\delta$ in Sec 3.1. Then, according to Alg.1, the algorithm converges after
> $$L_1^2 R^2 + \bar{k} + \frac{L_0 R}{\sqrt{\varepsilon}}$$
> $$=L_1^2 R^2 + \min[\log \left(\frac{1}{L_1^2 R^2}\right), 1]) + \frac{L_0 R}{\sqrt{\varepsilon}}$$
> oracle calls (up to constant factors) to an $\varepsilon$-solution for our choice of $\delta$ (see (7)). So, the term $\bar k$ never dominates.

---

> > ### Comment · Reviewer_XLXR · 2025-11-22
> >
> > Thank you for the response.
> >
> > > Yes, see our example in Section E.3.1.
> >
> > Now all experiments used Alg. 2, which requires $\psi^{-1}$. It is still not so clear how to incorporate the algorithm in practice when $\psi$ is not strictly increasing. A concrete numerical example with non-increasing $\psi$ using Alg.1 would be helpful.

---

> ### Author Response · Authors · 2025-11-22
>
> > Now all experiments used Alg. 2, which requires $\psi^{-1}$. It is still not so clear how to incorporate the algorithm in practice when $\psi$ is not strictly increasing. A concrete numerical example with non-increasing $\psi$ using Alg.1 would be helpful.
>
> Please consider the updated PDF, where we added Section A.4. In this section, we consider Alg. 1 with a $(3, L_0, L_1)$–smooth function and explain how to use Alg. 1 for a function with a non-monotonic $\psi$. Notice that $\psi$ is not needed in Alg. 1., and only used in the derivations of $\delta$ in (52).

---

### Official Review · Reviewer_4QVg · 2025-11-01

**Soundness:** 3
**Presentation:** 3
**Contribution:** 2
**Rating:** 4
**Confidence:** 2

**Summary:**

This paper proposes accelerated gradient methods under $(L_0, L_1)$-smoothness condition. It presents two algorithms, one with a brief GD warm start and another that adapts step sizes without a warm start, and establish the best-known oracle complexity $\sqrt{l(0)}R/\sqrt{\epsilon}$ in the small $\varepsilon$ regime. The analysis also extends to the generalized $(L_0, L_1)$-smoothness setting.

**Strengths:**

They proposed algorithm which established the best-known oracle complexity in the small $\epsilon$ regime with tailored Lyapunov function. The results align with optimal complexity under $l$-smoothness condition and are empirically validated on a toy problem. The proof sketch is properly presented and Table 1 clearly exhibits the contribution.

**Weaknesses:**

To be honest, I’m uncertain that the paper’s contribution meets the bar for acceptance at this venue. While the paper establishes a best-known bound, the guarantee is confined to the small $\epsilon$ regime, and the constant factor improvement over Li et al. (2024a) seems somewhat incremental.

**Questions:**

Is a lower-bound result established under this $(L_0, L_1)$-smoothness condition?

What are the technical challenges in extending the analysis to arbitrary $\epsilon$?

**Details Of Ethics Concerns:**

There is no ethics concerns.

---

> ### Author Response · Authors · 2025-11-13
>
> Thank you. Let us respond to the weaknesses.
>
> > To be honest, I’m uncertain that the paper’s contribution meets the bar for acceptance at this venue. While the paper establishes a best-known bound, the guarantee is confined to the small  regime, and ..
>
> **Notice that his work provides a new state-of-the-art theoretical complexity for modern and timely problems under generalized and $(L_0, L_1)$–smoothness. This direction is highly relevant to the community. Moreover, we have developed new proof techniques in Section 3.3 (that might be overlooked). For the ICLR optimization community, we strongly believe that these contributions are highly relevant.**
>
> >  ...the constant factor improvement over Li et al. (2024a) seems somewhat incremental.
>
> The difference can be huge. In the case of $(L_0, L_1)$–smoothness, the ratio is
> $\frac{\ell(||\nabla f(x_0)||)}{\ell(0)} = \frac{L_0 + L_1 ||\nabla f(x_0)||}{L_0}.$
> The norm can be as large as $\Theta(e^{L_1 R})$ for the function $e^{L_1 R}$. Thus, the difference is exponential!
>
> > Is a lower-bound result established under this (L_0, L_1)-smoothness condition?
>
> The lower bound was obtained in Section 2.1.2 of (Nesterov, 2018). Yurii Nesterov provided an example of a function that is $(L_0, 0)$–smooth, and therefore $(L_0, L_1)$–smooth for any $L_1 \ge 0$. Thus, we can use this lower bound to justify that our approach is optimal in the small-$\varepsilon$ regime (e.g., $\varepsilon \leq L_0 / L_1^4 R^2$).
>
> > What are the technical challenges in extending the analysis to arbitrary $\varepsilon$?
>
> Our analysis works for all $\varepsilon > 0,$ but we proved the optimality of our method only for small $\varepsilon.$
>
> ---
>
> Thank you for the questions. If you have more, please let us know.

---

> ### Comment · Reviewer_4QVg · 2025-11-21
>
> Thanks to the authors for their response.
>
>
>  (i) Regarding the second comment, if I understand correctly, $L_1$ and $R$ are constants, and hence $e^{L_1 R}$ is also a constant. Do the authors mean that the difference in constants is exponential?
>
> (ii) Also regarding the third comment: Nesterov's lower bound is originally stated for the $(L_0, 0)$ case. Is there no improved lower bound that adapts to the $(L_0, L_1)$ setup, i.e., that uses the information about $L_1$ to obtain a tighter bound?

---

> ### Author Response · Authors · 2025-11-21
>
> Thank you for the response.
>
> > (i) Regarding the second comment, if I understand correctly, $L_1$ and $R$ are constants, and hence $e^{L_1 R}$ is also a constant. Do the authors mean that the difference in constants is exponential?
>
> The parameters $L_1$ and $R$ are parameters of the function. They are not universal constants (like 2 or 7); they can be arbitrarily large and depend on the function $f$ and the initial point $x^0$. Thus, $e^{L_1 R}$ can be very large. In the context of $(L_0, L_1)$-smoothness, one of the main theoretical goals is to develop a method with the best possible dependence on $L_0$, $L_1$, and $R$ with respect to the $\varepsilon$ accuracy. We have achieved a new milestone and developed methods that attain an oracle complexity of $\mathcal{O}\left(\sqrt{\frac{L_0 R^2}{\varepsilon}}\right)$.
>
> > Also regarding the third comment: Nesterov's lower bound is originally stated for the $(L_0, 0)$ case. Is there no improved lower bound that adapts to the $(L_0, L_1)$ setup, i.e., that uses the information about $L_1$ to obtain a tighter bound?
>
> As far as we know, the answer is no. We only know for sure that the lower bound is $\Omega\left(\sqrt{\frac{L_0 R^2}{\varepsilon}}\right).$ **Exactly what we achieve for small $\varepsilon$! Even achieving this lower bound was an open problem!**

---

> ### Comment · Reviewer_4QVg · 2025-11-21
>
> Thank you to the authors for their response. However, since I am still not sure the contribution meets the bar, I will maintain my score.

---

### Official Review · Reviewer_2a6J · 2025-11-01

**Soundness:** 2
**Presentation:** 2
**Contribution:** 2
**Rating:** 4
**Confidence:** 4

**Summary:**

This paper considers problems that satisfy $\ell$-smoothness conditions, which genearlize both standard and $(L_0, L_1)$-smoothness. For this setting, they achieve a $O(\sqrt{\ell(0)}R/\sqrt{\varepsilon}) + L_1^2R^2$ first-order oracle complexity to find an $\epsilon$-approximate solution.

**Strengths:**

The primary strength of the paper is that it improves the previous $\ell(||\nabla f(x^0)||)$ dependence to $\ell(0)$ (up to considerations of additive terms, discussed below in "Weaknesses"), and this helps better place the result in the context of classic lower bound in smooth convex optimization.

**Weaknesses:**

One issue is that the algorithm needs $\Gamma_0$, $\bar{R}$. Do the other algorithms in Table 1 require these? If not, then the results are not directly comparable, and it would then be important to explain these caveats as an additional part of the table. The authors claim (erroneously) the complexity is optimal (line 250). The authors should specify the range of $\varepsilon$ where they claim optimality, and should emphasize the additive term which prevents them from actually being optimal. The additive term is $L\_1^2R^2$, which can dominate for some ranges of $\varepsilon$, $L\_1$, $ R $, $L\_0$. This needs clarifying for the exact range of improvements, and to point out when previous works dominate this work, for proper comparison.

Following this, the work could benefit from providing a clearer description of why this result is important in the face of previous work, since the overall improvement seems quite slight in that it only affects the smoothness parameters (and furthermore at the cost of a potentially worse additive term in some cases), and the techniques resemble those in Vankov et al. Because of these concerns, there is some hesitance felt about whether these results are significant enough to warrant acceptance, especially in view of the caveats above yet to be completely addressed.

**Questions:**

What precisely is $\nu$ in Table 1? (Can its dependence on parameters of $f$ be elaborated on?)

---

> ### Author Response · Authors · 2025-11-13
>
> Thank you. **Notice that we've updated and incorporated the required changes into the PDF.**
>
> > One issue is that the algorithm needs $\Gamma_0$, $\bar{R}$. Do the other algorithms in Table 1 require these? If not, then the results are not directly comparable, and it would then be important to explain these caveats as an additional part of the table.
>
> We agree that different methods provide different trade-off. We have added a new column to Table 1 with required parameters for each method. For instance, AGD by (Gorbunov et al.) requires fewer parameters, but it has an exponential dependence on them. On the other hand, our method requires four parameters (and is semi-adaptive to two of them), but it is optimal for small $\varepsilon$.
>
> > The authors claim (erroneously) the complexity is optimal (line 250).
>
> We say that it is optimal for small $\varepsilon$. We are cautious with our claims and add “for small $\varepsilon$” everywhere it is necessary. We also state that our result is optimal in the small-$\varepsilon$ regime (Line 253). For clarity, we have duplicated this information in Line 250 to avoid confusion.
>
> > The authors should specify the range of $\varepsilon$ where they claim optimality, and should emphasize the additive term which prevents them from actually being optimal. The additive term is $L_1^2R^2$, which can dominate for some ranges of $\varepsilon$, $L_1$, $ R $, $L_0$. This needs clarifying for the exact range of improvements, and to point out when previous works dominate this work, for proper comparison.
>
> The regime where our algorithm is optimal seems straightforward: since the complexity is $\frac{\sqrt{L_0} R}{\sqrt{\varepsilon}} + L_1^2 R^2$, the optimality and strict improvement, compared to other methods, are achieved for all parameters such that $\frac{\sqrt{L_0} R}{\sqrt{\varepsilon}} \ge L_1^2 R^2$. Notice that we obtain an even better non-dominant term in Section 4.1.
>
> > Following this, the work could benefit from providing a clearer description of why this result is important in the face of previous work, since the overall improvement seems quite slight in that it only affects the smoothness parameters (and furthermore at the cost of a potentially worse additive term in some cases), and the techniques resemble those in Vankov et al.
>
> **This work provides a new state-of-the-art theoretical complexity for modern and timely problems under generalized and $(L_0, L_1)$–smoothness. This direction is highly relevant to the community. Moreover, we have developed new proof techniques in Section 3.3 (that might be overlooked). For the ICLR optimization community, we strongly believe that these contributions are highly relevant.**
>
> > What precisely is $\eta$ in Table 1?
>
> In [1], this quantity is the number of oracle calls required to solve an auxiliary problem. This value is not specified in the original paper; however, it clearly depends on $f$ (and on the desired accuracy), since the auxiliary problem itself depends on $f$. Notice that our method computes one gradient per iteration, whereas [1] requires solving the auxiliary problem. Thus, their approach is slower from both practical and theoretical perspectives.
>
> ---
>
> **We believe that we have addressed all the concerns, and hope that the reviewer will reconsider the score. We are happy to respond to any further questions.**
>
> [1]: Daniil Vankov, Anton Rodomanov, Angelia Nedich, Lalitha Sankar, and Sebastian U Stich. Optimizing (L0,L1)-smooth functions by gradient methods. arXiv preprint arXiv:2410.10800, 2024

---

### Official Review · Reviewer_fCE5 · 2025-11-02

**Soundness:** 2
**Presentation:** 3
**Contribution:** 3
**Rating:** 4
**Confidence:** 4

**Summary:**

This paper proposes two new Accelerated GD variants designed to optimize $\ell$-smooth functions (the generalized version of $L$-smoothness and $(L_0, L_1)$-smoothness). The authors provide convergence guarantees for their proposed methods, showing that in the small accuracy regime their methods attain near-optimal convergence of $\mathcal{O}(\sqrt{\ell(0)}R/\sqrt{\varepsilon})$, where $R =\\|x_0 - x^{\star}\\|$, thus imporving upon prior approaches. Between the two of them, these algorithms cover both the case of sub- and superquadratic $\ell$. Preliminary experiments are provided.

**Strengths:**

The topic is relevant to the research community, and the paper is well-structured and well-written. To the best of my knowledge, the related literature is appropriately covered, and significant parts of the technical approach are novel. The contribution is significant for both theory and practice, since it helps delineate the reach/limitations of classical methods under generalized smoothness, and can provide practitioners in, e.g., scientific computing fields, with potentially improved tools.

**Weaknesses:**

1. **Technical approach**
	* Assumption 2.3 states that "$ f : \mathbb{R}^d \to \mathbb{R} \cup \{\infty\} $ [...] attains its minimum at a (non-unique)
$x^\ast \in \mathbb{R}^d $ [...]". However, none of the motivating examples in line 051 do satisfy this over $\mathbb{R}^d$ (for the case of $x^p$, consider $p$-odd).
	* The method addresses contrained optimization, yet the proof of Lemma B.3 uses the result of Lemma B.1 by replacing $\nabla f(y)$ with $\nabla f (x^\star)$ which is set to zero. The constrained optimum $x^\star$ does not necessarily satisfy $\nabla f (x^\star) = 0$, so the result is problematic. Could you please address this and the possible ramifications of it?

2. **Presentation**
	* A comparison between the stepsize's dependence on problem constants of Alg2 vs. [1,2,3] is missing, and would help in understanding the tradeoffs between this and previous methods.

3. **Experiments**
	* The experiments are too simplistic, and only compare Algorithm 2 with Tyurin's GD variant (unaccelerated). The method should be compared with the Accelerated versions of [1, 2, 3]. For a fair comparison where auxiliary subroutines are concerned, convergence in terms of wall clock time should be considered. This is useful for understanding the practical behaviours of these methods relative to each other, since it is likely that despite the worse convergence upper bounds, the prior algorithms are still competitive in practice in the small $\varepsilon$ regime.
	* Experiments with various degrees of overestimation for $\bar{R}$ and $\Gamma_0$ should be conducted to understand Alg. 2's sensitivity to hyperparameter tuning (even if the $\varepsilon$-dependent term only depends on $R$, and not $\bar{R}$).
	* Less pressing: ideally, an experiment should be included on a practically-relevant (empirically determined) $(L_0, L_1)$-loss, in order to assess the method's sensitivity to tuning in practice


[1] Haochuan Li, Jian Qian, Yi Tian, Alexander Rakhlin, and Ali Jadbabaie. Convex and non-convex optimization under generalized smoothness. Advances in Neural Information Processing Systems, 36, 2024a.

[2] Eduard Gorbunov, Nazarii Tupitsa, Sayantan Choudhury, Alen Aliev, Peter Richt´arik, Samuel Horv´ath, and Martin Tak´aˇc. Methods for convex (L0,L1)-smooth optimization: Clipping, acceleration, and adaptivity. In International Conference on Learning Representations, 2025.

[3] Daniil Vankov, Anton Rodomanov, Angelia Nedich, Lalitha Sankar, and Sebastian U Stich. Optimizing (L0,L1)-smooth functions by gradient methods. arXiv preprint arXiv:2410.10800, 2024.

**Questions:**

Please see comments above.

---

> ### Author Response · Authors · 2025-11-13
>
> Thank you. **Notice that we've updated and incorporated the required changes into the PDF.**
>
> > **Technical approach**
>
>    a. These are motivating examples for generalized smoothness. For example, a function that is not $L$-smooth yet has a finite minimum is $-\mu x + e^x$, or the function from Section A.1.
>
>    b. There might be some confusion. We consider unconstrained optimization; however, the domain of the function, where it is finite, can be a strict open subset of $\mathbb{R}^d$. Since $X$ is open and $x^* \in X$, one can easily show that $\nabla f(x^*) = 0$, and there is no contradiction.
>
> > **Presentation**
>
> We've added one more column to Table 1. Different methods provide different trade-offs. For instance, AGD by (Gorbunov et al.) requires fewer parameters, but it has an exponential dependence on them. On the other hand, our method requires four parameters (and is semi-adaptive to two of them), but it is optimal for small $\varepsilon$.
>
> > **Experiments**
>
> We have added an experiment to Section A where we compare other AGD methods with ours, and we also include a sensitivity-to-parameters study. All experiments are consistent with the theoretical results.
>
> ---
>
> **We believe that we have addressed all the concerns:** all questions have been clarified, the dependencies of each method have been added to Table 1, and the additional experiments with other methods and the sensitivity to parameter choices have been conducted. **Considering this, we hope that the reviewer will reconsider the score. We are happy to respond to any further questions.**

---

### Author Response · Authors · 2025-12-01
**Public Summary Comment to AC, Reviewers, and Community**

Dear AC, reviewers, and the ICLR optimization community,

Thank you for your time and effort. In this comment, we want to summarize the review and rebuttal process and emphasize the importance of our work.

**(Introduction):** We study optimization problems under $(L_{0},L_{1})$-smoothness, $||\nabla^2 f(x)|| \leq L_0 + L_1 ||\nabla f(x)||,$ and generalized smoothness, $||\nabla^{2}f(x)|| \le \ell\left(||\nabla f(x)||\right).$ These are important and timely assumptions in the modern optimization community, where the main challenge is to design an efficient accelerated method with the best possible computational complexities. It turns out that this is a very challenging task. The first work that designed an accelerated method is [1]. They developed a method with the oracle complexity $O\left(\frac{\sqrt{L_0 + {\color{red} L_1 ||\nabla f(x^0)||}} R}{\sqrt{\varepsilon}}\right).$ Notice that their complexity depends on $||\nabla f(x^0)||,$ which can be exponentially large in the parameters $L_1$ and $R := ||x^0 - x^*||.$ Subsequently, two other concurrent works [2] and [3] developed two other approaches with the complexities ${\color{red}\exp(L_1 R)} \times \frac{\sqrt{L_0} R}{\sqrt{\varepsilon}}$ and ${\color{red} \nu} \times \frac{\sqrt{L_0} R}{\sqrt{\varepsilon}},$ where $\nu$ is not a universal constant. However, these complexities are still suboptimal w.r.t. $\varepsilon.$

**(Our contribution):** At the same time, we know that the lower bound is $\Omega\left(\frac{\sqrt{L_0} R}{\sqrt{\varepsilon}}\right)$ w.r.t. accuracy $\varepsilon.$ It turns out the problem of designing a method that would achieve the lower bound $\Omega\left(\frac{\sqrt{L_0} R}{\sqrt{\varepsilon}}\right)$ under these assumptions is a very challenging and non-trivial task. **At least three different works [1-3] were not able to achieve this! In our work, finally, we consider an accelerated method and prove that this method can achieve this lower bound. For mathematicians and the ICLR optimization community, this is a very important milestone.**

---

We have read all the reviews and believe that we addressed all the concerns: i) Reviewers fCE5 and XLXR asked for additional experiments; we added them to Section A. ii) Reviewers fCE5 and 2a6J asked to clarify the dependence of each method on input parameters; this is also added to Table 1. iii) all other comments are clarified and, where necessary, are incorporated into the updated pdf.

In total, we are sure that all comments by Reviewers fCE5 and XLXR are addressed. However, what is left are the comments by Reviewers 2a6J, 4QVg , and MkRn, regarding the significance of our contribution:

> Reviewers 2a6J: "Following this, the work could benefit from providing a clearer description of why this result is important in the face of previous work, since the overall improvement seems quite slight in that it only affects the smoothness parameters [...] Because of these concerns, there is some hesitance felt about whether these results are significant enough to warrant acceptance, especially in view of the caveats above yet to be completely addressed."

> Reviewer 4QVg: "To be honest, I’m uncertain that the paper’s contribution meets the bar for acceptance at this venue. [...]"

> Reviewer MkRn: "The proof technique is not novel.  [...]"

**We respectfully disagree with these comments. This work provides a new state-of-the-art theoretical complexity for modern and timely problems under generalized and $(L_0, L_1)$–smoothness. This direction is highly relevant to the community [1-3], and getting the best possible complexity is the central question in optimization. Developing new proof techniques (see Section 3.3 that might be overlooked), we achieved the optimal oracle complexity $\frac{\sqrt{L_0} R}{\sqrt{\varepsilon}}$ w.r.t. $\varepsilon.$ For the ICLR optimization community, we strongly believe that this is a very important result.**

---

Thank you once again! We hope that this comment clarifies our work and contributions.

Authors

---

[1]: Haochuan Li, Jian Qian, Yi Tian, Alexander Rakhlin, and Ali Jadbabaie. Convex and non-convex optimization under generalized smoothness. (NeurIPS 2024)

[2]: Eduard Gorbunov, Nazarii Tupitsa, Sayantan Choudhury, Alen Aliev, Peter Richtarik, Samuel Horvath,
and Martin Takac. Methods for convex (L0,L1)-smooth optimization: Clipping, acceleration, and adaptivity. (ICLR 2025)

[3]: Daniil Vankov, Anton Rodomanov, Angelia Nedich, Lalitha Sankar, and Sebastian U Stich. Optimizing (L0,L1)-smooth functions by gradient methods (ICLR 2025)

---

### Meta-Review · Area_Chair_P5Pw · 2025-12-30

**Summary:**

This paper studies accelerated first-order methods for convex optimization under generalized $\ell$-smoothness and $(L_0, L_1)$-smoothness. It addresses an important open question: whether accelerated gradient methods can achieve oracle complexity matching the classical lower bounds in the small-$\varepsilon$ regime under these generalized smoothness assumptions. The authors propose new accelerated algorithms and develop a detailed theoretical analysis, claiming rates of $O(\sqrt{\ell(0)}R/\sqrt{\varepsilon})$ in the general $\ell$-smooth setting and $O(\sqrt{L_0}R/\sqrt{\varepsilon})$ for $(L_0,L_1)$-smooth functions, which are optimal in this regime.

Overall, the problem addressed is timely and relevant to the optimization community, and the paper demonstrates substantial technical effort. However, despite these strengths, none of the reviewers ultimately support acceptance, citing concerns about the clarity, positioning, and perceived incremental nature of the contribution relative to recent closely related work.

While I personally find the problem addressed by the paper important and agree with the authors that achieving optimal accelerated rates under generalized smoothness is a meaningful theoretical goal, the current version does not sufficiently highlight and communicate its core algorithmic and theoretical contributions relative to prior work. In light of the uniformly cautious or negative reviewer recommendations, I recommend rejecting the manuscript.

I would strongly encourage the authors to revise the presentation in a future submission to more clearly emphasize:

* The precise algorithmic and proof-level novelties,
* The regimes in which the proposed methods are strictly preferable,
* And the broader significance of these results for the optimization community.

**Reviewer Concerns:**

The main concerns raised by the reviewers can be summarized as follows:

* **Significance and novelty:** Several reviewers questioned whether the improvement over prior accelerated methods is sufficiently substantial, noting that the gains primarily affect smoothness-dependent constants and hold only in the small-(\varepsilon) regime.
* **Positioning relative to prior work:** Reviewers found that the paper does not clearly and convincingly articulate how the proposed algorithms and proof techniques fundamentally differ from, or improve upon, recent concurrent approaches, especially those with similar accelerated frameworks.
* **Clarity of optimality claims:** There was confusion regarding the scope of the optimality statements, particularly the role of additive terms that can dominate outside the small-(\varepsilon) regime.
* **Presentation and comparisons:** Reviewers requested clearer explanations of parameter requirements, more transparent comparisons in Table 1, and stronger justification for why the proposed methods should be preferred in practice or theory over existing alternatives.

**Reviewer Scores:**

In the rebuttal and revised manuscript, the authors made a good-faith effort to address reviewer feedback. In particular, they:

* Clarified that optimality claims are restricted to the small-(\varepsilon) regime.
* Expanded Table 1 to explicitly list required parameters for competing methods.
* Added additional experiments and sensitivity analyses to address empirical concerns.
* Provided further discussion of the proof techniques and their differences from prior work.

These changes improve the clarity and completeness of the submission. However, even after the rebuttal, several reviewers remained unconvinced that the contribution rises above incremental improvements, especially given the similarity to existing accelerated schemes and the narrow regime in which the advantages are most pronounced.

---

### Decision · Program_Chairs · 2026-01-26

Reject